# The tipping points and early-warning indicators for Pine Island Glacier, West Antarctica

Sebastian H. R. Rosier[1*], Ronja Reese[2], Jonathan F. Donges[2,3], Jan De Rydt[1], G. Hilmar Gudmundsson[1], Ricarda Winkelmann[2,4]

[1] Department of Geography and Environmental Sciences, Northumbria University, Newcastle, UK
[2] Earth System Analysis, Potsdam Institute for Climate Impact Research (PIK), Member of the Leibniz Association, P.O. Box 60 12 03, 14412 Potsdam, Germany
[3] Stockholm Resilience Centre, Stockholm University, Kräftriket 2B, SE-10691 Stockholm, Sweden
[4] University of Potsdam, Institute of Physics and Astronomy, Karl-Liebknecht-Str. 24–25, 14476 Potsdam, Germany

*Correspondence to*: Sebastian Rosier (Sebastian.rosier@northumbria.ac.uk)

**Abstract.** Mass loss from the Antarctic Ice Sheet is the main source of uncertainty in projections of future sea-level rise, with important implications for coastal regions worldwide. Central to ongoing and future changes is the marine ice sheet instability: once a critical threshold, or tipping point, is crossed, ice-internal dynamics can drive a self-sustaining retreat committing a glacier to irreversible, rapid and substantial ice loss. This process might have already been triggered in the Amundsen Sea region, where Pine Island and Thwaites glaciers dominate the current mass loss from Antarctica, but modelling and observational techniques have not been able to establish this rigorously, leading to divergent views on the future mass loss of the WAIS. Here, we aim at closing this knowledge gap by conducting a systematic investigation of the stability regime of Pine Island Glacier. To this end we show that early warning indicators in model simulations robustly detect the onset of the marine ice sheet instability. We are thereby able to identify three distinct tipping points in response to increases in ocean-induced melt. The third and final event, triggered by an ocean warming of approximately 1.2 °C from the steady state model configuration, leads to a retreat of the entire glacier that could initiate a collapse of the West Antarctic Ice Sheet.

## 1. Introduction

The West Antarctic Ice Sheet (WAIS) is a tipping element in the earth's climate system; a major component of the earth system susceptible to tipping point behaviour (Lenton *et al.*, 2008). Its collapse, potentially driven by the Marine Ice Sheet Instability (MISI; Feldmann and Levermann, 2015), would result in over 3m of sea level rise (Fretwell *et al.*, 2013). Key to the MISI are the conditions at the grounding line - the transition across which grounded ice begins to float on the ocean forming ice shelves. In steady state, ice flux across the grounding line balances the surface accumulation upstream. If grounding line retreat causes grounding line flux to increase and this is not balanced by a corresponding increase in accumulation, the net mass balance is negative and retreat will continue (Weertman, 1974; Schoof, 2007). Conversely, grounding line advance leading to an increasing accumulation greater than the change in flux will lead to a continued advance. In this regime, a small perturbation in the system can result in the system crossing a tipping point, beyond which a positive feedback propels the system to a

contrasting state (Fig. 1c). A complex range of factors can either cause or suppress the MISI (Haseloff, 2018; Pegler, 2018, O'Leary *et al.*, 2013; Gomez *et al.* 2010; Robel *et al.* 2016) and the difficulties in predicting this behaviour are a major source of uncertainty for future sea level rise projections (Church *et al.*, 2013; Bamber *et al.*, 2019; Oppenheimer *et al.*, in press; 35    Robel *et al.* 2019).

One area of particular concern is the Amundsen Sea region. Pine Island (PIG) and Thwaites glaciers, the two largest glaciers in the area, are believed to be particularly vulnerable to the MISI (Favier *et al.*, 2014; Rignot *et al.*, 2014). Palaeo and observational records of PIG show a history of retreat, driven by both natural and anthropogenic variability in ocean forcing 40    (Jenkins *et al.*, 2018; Holland *et al.*, 2019). One possible MISI driven retreat might have happened when PIG unpinned from a submarine ridge in the 1940s (Jenkins *et al.,* 2010; Smith *et al.*, 2016). Recent modelling studies indicate that a larger scale MISI event may now be underway for both Pine Island and Thwaites glaciers that would lead to substantial and sustained mass loss throughout the coming centuries (Favier *et al.*, 2014; Jenkins *et al.*, 2016; Joughin *et al.,* 2010). Being able to identify a MISI driven retreat and differentiate this from a retreat where a tipping point has not been crossed is vital information for 45    projections of future sea level rise. One of the major hurdles in determining whether a tipping point has been crossed is that currently this necessitates time consuming steady-state simulations to calculate the hysteresis behaviour of an identified period of retreat (e.g. Garbe *et al.* 2020). An alternative methodology that can be applied directly to transient simulations as a post-processing step would therefore be useful to the ice-sheet modelling community. Such tools based on early-warning indicators are presented in this paper.

50

The tipping behaviour of the MISI is an example of a saddle-node (or fold) bifurcation in which three equilibria exist; an upper and lower stable branch and a middle unstable branch (Fig. 1c; Schoof, 2012). Starting on the upper stable branch, perturbing the system beyond a tipping point ($x_1$ in Fig. 1c) will induce a qualitative shift to the lower and contrasting stable state. Importantly (and in contrast to a system such as that shown in Fig.1a and 1b), in order to restore conditions to the state prior 55    to a collapse it is not sufficient to simply reverse the forcing to its previous value. Instead, the forcing must be taken back further (to point $x_2$), which in some cases may be far beyond the parameter range that triggered the initial collapse. This type of behaviour is known as hysteresis. A large change in response to a small forcing is not necessarily indicative of a hysteresis, as shown in Fig.1b. Tipping points are crossed in both Fig. 1c and Fig. 1d and both cases are often referred to as irreversible, although the two are distinct in that only Fig. 1d is irreversible for any change in the tested range of the control parameter. 60    Hereafter we will refer to the former as irreversible, in line with previous studies, and the latter as permanently irreversible, to differentiate the two. Diagnosing whether a tipping point has been crossed without some prior knowledge of the system is not generally possible without reversing the forcing to see if a hysteresis has occurred. An alternative approach to identify tipping points is based on a process known as *critical slowing down,* which is known to precede saddle-node bifurcations of this type (Wissel, 1984; van Nes and Scheffer, 2007; Davos *et al.*, 2008; Scheffer *et al.*, 2009). Critical slowing down is a general feature 65    of non-linear systems and refers to an increase in the time a system takes to recover from perturbations as a tipping point is

approached (Wissel, 1984). We will explore both hysteresis and critical slowing down as indicators of tipping points in our model simulations.

In Section 2, we explain critical slowing down and early warning indicators in the context of the MISI. We then map out the stability regime of Pine Island Glacier (PIG) using numerical model simulations. We force the model with a slowly increasing ocean melt rate and identify three periods of rapid retreat with the methodology explained in Sect. 3.1. Using statistical tools from dynamical systems theory we find critical slowing down preceding each of these retreat events and go on to demonstrate that these are indeed tipping points in Sect. 4. This is confirmed by analysing the hysteresis behaviour of the glacier, showing the existence of unstable grounding line positions. To our knowledge, this is the first time that the stability regime of PIG has been investigated in this detail and the first time that tipping point indicators have been applied to ice sheet model simulations. Our results reveal the existence of multiple smaller tipping points that when crossed could easily be mis-identified as simply periods of rapid retreat, with the irreversible and the self-sustained aspect of the retreat being missed.

## 2. Critical slowing down and early warning indicators

As certain classes of complex systems approach a tipping point, they show early warning signals which can allow us to anticipate or even predict the onset of a tipping event by means of early warning indicators (EWIs; Wissel, 1984). Early warning signals have been found to precede, for example, collapse of the thermohaline circulation (Held and Kleinen, 2014; Lenton, 2011), onset of epileptic seizures (Litt, 2001; McSharry and Tarassenko, 2003), crashes in financial markets (May *et al.*, 2008; Diks *et al.*, 2018), onset of glacial terminations (Lenton, 2011) and wildlife population collapses (Scheffer *et al.*, 2001). Although most commonly used to detect the onset of saddle-node bifurcations, of which the MISI is an example, they are not strictly limited to bifurcations of this type and have, for example, also been successfully used to indicate the onset of Hopf bifurcations (Chisholm & Filotas, 2009).

## 2.1 Critical slowing down preceding the Marine Ice Sheet Instability

Critical slowing down is one example of an early warning signal that has been used in the past for both model output and observational records such as paleoclimate data, with the aim of detecting an approaching bifurcation (Held and Kleinen, 2004; Livina and Lenton, 2007; Dakos *et al.* 2008; Lenton *et al.* 2009; Lenton *et al.* 2012b). Critical slowing down is so called because, as a non-linear system is gradually forced towards a bifurcation, that system will become more 'sluggish' in its response to perturbations (see middle panel of Fig. 2). This can be shown mathematically, because the dominant eigenvalue of the system tends to zero as a bifurcation point is approached (Wissel, 1984), or, equivalently, the recovery time (i.e. the time it takes for a system to return to a steady state after small perturbations) tends to infinity. The response time of a glacier to external forcing has also been shown analytically to increase as a MISI bifurcation is approached (Robel *et al.* 2018). While

critical slowing down is a general behaviour of the dynamics underlying the MISI, the question remains whether it can be reliably detected in the context of a complex glacier where many other processes are at play.

As a first step to addressing this question, we model a MISI in an idealised flowline setup of a marine ice sheet. In this setup, we determine the change in recovery time before a tipping point directly through multiple stepwise perturbations of the control parameter (Appendix A). Our setup closely resembles the MISMIP experiments (Pattyn *et al.*, 2012) and indeed hints of critical slowing down can be identified in that paper (Fig. 2 in Pattyn *et al.* 2012). The results in Appendix A show that critical slowing down is easily identified preceding both MISI driven advance and retreat bifurcations.  This demonstrates that there is at least

the potential that critical slowing down could be found in a less simplified modelling framework. This is not a priori clear and, for example, adding noise to the bed topography reduces the ability to identify early warning, as detailed in the Appendix. Identifying critical slowing down in this stepwise perturbation manner is appealing because it directly extracts the change in response time that we are searching for, however it is not practical for a realistic model forcing which would not normally take the form of a step function. A more general approach, which we adopt for our simulation of PIG, is to use EWIs to analyse the

recovery time of the system as it is forced with natural variability.

## 2.2. Early warning indicators

As the field of EWIs has expanded, more methods have been developed for extracting critical slowing down information from model results and observational records. These methods seek to approximate the system recovery time from some measure of the system state. The challenge is that, for most real-world applications, natural forcing does not take the form of a step function

and the system is continuously perturbed and so cannot return to a true steady state. However, if the recovery time of a system is indeed increasing, the response to a continual stochastic forcing could be detected as a tendency for each measurement of the system state to be more similar to the previous measurement, sometimes referred to as an increase in "memory" of small perturbations. This is shown conceptually and with examples extracted from our PIG model in Fig. 2. One common way to measure this effect is by sampling the data at discrete time intervals and calculating the lag-1 autocorrelation, i.e. the correlation

between values that are one time interval apart (examples given in Fig. 2). This measure, which we refer to hereafter as the *ACF indicator*, should increase as a tipping point is approached (Dakos *et al.*, 2008; Ives, 1995). Since recovery time tends to infinity as the bifurcation is approached, successive system states should become more and more similar and the ACF indicator should tend to one. An alternative measure that also seeks to identify changes in recovery time is to use the Detrended Fluctuation Analysis algorithm (Livina and Lenton, 2007; Lenton *et al.*, 2012a; Lenton *et al.*, 2012b). This first calculates the

mean-centered cumulative sum of the time series, splits the result into epochs of length $n$ which are detrended and then calculates the RMS $F(n)$ for each epoch. This is repeated for epochs of different length and finally an exponent $\alpha$ can be fitted in log-log space such that $F(n) \propto n^{\alpha}$.  This exponent yields information on the self-correlation of the original time series, whereby a value of 0.5 corresponds to uncorrelated white noise and greater values indicate increasing "memory" up to a

maximum of 1.5. To aid comparison with the ACF indicator, we rescale the exponent so that it reaches a critical value of 1 and call this the *DFA indicator* (Livina and Lenton, 2007). These indicators can be supported by analysing the variance of the system state. Variance can be shown to increase as a tipping point is approached, since perturbations to the system decay more slowly and thus large shifts from the mean state will persist for longer (Scheffer *et al.* 2009). No critical value exists in this case, but a persistent positive trend in variance serves as additional evidence that a tipping point is being approached.

## 3. Methods

We conduct a quasi-steady modelling experiment whereby we subject PIG to slowly increasing rates of basal melt underneath its adjacent ice shelf (Fig. 3). Conducting a transient simulation with an evolving basal melt that exactly tracks the equilibrium curve (Fig. 1c) is not computationally feasible or necessary for our purposes. Thus, we adopt this quasi-steady modelling approach in which the forcing increases slowly enough that it approximates the steady state behaviour, but faster than the long response timescales of the glacier would require to be truly in equilibrium. Quasi-steady state experiments have previously been successfully applied to identify the tipping point of the Greenland Ice Sheet with respect to the melt-elevation feedback (Robinson *et al.,* 2012) and to identify hysteresis of the Antarctic Ice Sheet (Garbe *et al.*, 2020). In Garbe *et al.* (2020) it was shown that such transient experiments enable identification of hysteresis behaviour, while the exact shape of the curve must be mapped out with equilibrium simulations. We accompany the quasi-steady simulations with simulations that run to a true steady state for constant values of the control parameter at discrete values (these simulations continue until the change in ice volume is approximately equal to zero). We use basal melt rate as the control parameter, i.e. the parameter that we will change to drive the system towards a tipping point. We make this choice since erosion of ice shelves by the intrusion of warm ocean currents is widely accepted as the mechanism responsible for the considerable changes currently observed in this region (Shepherd *et al.,* 2004; Rignot *et al.*, 2014; Rignot 1998; Joughin *et al.*, 2010; Park *et al.*, 2013; Gudmundsson *et al.,* 2019). Sub-ice-shelf melt rates are increased linearly (with additional variability as explained below) from a value that generates a steady state for the present-day glacier configuration. Based on the numerical experiments we then evaluate EWIs to test for critical slowing down.

### 3.1 Model description

All simulations use the community Úa ice-flow model (Gudmundsson *et al.,* 2012; Gudmundsson 2013, Gudmundsson 2020), which solves the dynamical equations for ice flow in the shallow ice stream approximation (SSTREAM or SSA; Hutter, 1983). Bedrock geometry for the PIG domain is a combination of the R-Topo2 dataset (Schaffer *et al.,* 2016) and, where available, an updated bathymetry of the Amundsen Sea Embayment (Millan *et al.,* 2014). Surface ice topography is from CryoSat-2 altimetry (Slater *et al.,* 2018). Depth-averaged ice density is calculated using a meteoric ice density of $917\,\mathrm{kg\,m^{-3}}$ together with

firn depths obtained from the RACMO2.1 firn densification model (Ligtenberg *et al.,* 2011). Snow accumulation is a
climatological record obtained from RACMO2.1 and constant in time (Lenaerts *et al.,* 2012).

Viscous ice deformation is described by the Glen Steineman flow law $\dot{\varepsilon} = A\tau_E^n$ with exponent $n = 3$ and basal motion is
modelled using a Weertman sliding law $u_b = C\tau_b{}^m$ with exponent $m = 3$. The constituitive law and the sliding law use
spatially varying parameters for the ice rate factor $(A)$ and basal slipperiness $(C)$, respectively, to initialise the model with
present day ice velocities. These are obtained via optimization methods using satellite observations of surface ice velocity from
the Landsat 8 dataset (Scambos *et al.* 2016; Fahnestock *et al.* 2016). An optimal solution is obtained by minimising a cost
function that includes both the misfit between observed and modelled velocities and regularisation terms. An additional term
in the cost function penalises initial rates of ice thickness change in order to ensure that these are close to zero at the start of
simulations. This approach helps to provide a steady-state configuration of PIG from which we can conduct our perturbation
experiments.

The Úa model solves the system of equations with the finite element method on an unstructured mesh, generated with mesh2d
(Engwirda *et al.* 2014). The mesh remains fixed throughout the simulation to avoid contaminating the time series with errors
resulting from remapping fields onto a new mesh. The mesh is refined in regions of high strain rate gradients, fast ice flow and
around the grounding line. The region of grounding line mesh refinement, in which the average element size is ~750m, extends
upstream sufficiently far so that the grounding line always remains within this region until after the final MISI collapse.

Basal melt rates are calculated using a widely used, local quadratic dependency on thermal forcing:

$$M = f\gamma_T \left(\frac{\rho_w c_p}{\rho_i L_i}\right)^2 (T_0 - T_f)|T_0 - T_f|,$$

where $\gamma_T$ is the constant heat exchange velocity, $\rho_w$ is sea water density, $c_p$ is the specific heat capacity of water, $\rho_i$ is ice
density, $L_i$ is the latent heat of fusion of ice, $T_0$ is the thermal forcing and $T_f$ is the freezing temperature (Favier *et al.* 2019).
Melt rates are only applied beneath fully floating elements to ensure that no melting can possibly occur upstream of the
grounding line (Seroussi and Morlinghem, 2018). The initial melt rate factor $(f)$ is chosen such that the model finds a steady
state with a grounding line approximately coincident with its position as given in Bedmap2 (Fretwell *et al.* 2013). This melt
rate factor is the aforementioned control parameter that drives changes in the model, some of which may be identifiable as
tipping points.

To effectively extract information about the system's recovery time using the statistical methods outlined in Sect. 2, we need
to perturb the model in a way that has some measurable impact on the system state. A slow and monotonically increasing
forcing would make our chosen approach impractical and is arguably as unrealistic as a stepwise perturbation. We therefore
add natural variability to the linearly increasing melt rate factor $(f)$. There is strong evidence that the inferred and observed

changes of PIG over the last century can be linked to changes in thermocline depth of the Amundsen Sea shelf, which in turn is influenced by an atmospheric Rossby wave train originating in the Pacific Ocean (Jenkins *et al.* 2018). Following Jenkins *et al.* (2018), we use a ~130 year time series of central tropical pacific sea surface temperature anomaly as a proxy for relevant variability in our melt rate forcing. We create an autoregressive (AR) model-based surrogate from this time series using the Yule-Walker method to fit the AR model and minimum description length to determine the maximum order of the model. This new surrogate time series has the same decadal variability that would be expected for the melting beneath PIG and can be extended to any length required. As shown in more detail below, by superimposing this signal onto the linearly increasing melt rate factor we ensure that the system response contains sufficient variability to extract information about critical slowing down and thereby enable the calculation of EWIs. Furthermore, using natural variability enables us to test the versatility of EWIs if they were to be applied directly to observations.

## 3.2 Detecting critical slowing down

We have already established the control parameter for our model, but another important decision to make is what model output should be used as a measure of the system state. One choice could be changes in ice volume, since it can be related to sea level rise and ice sheet model simulations tend to focus on this result. However, ice volume varies very smoothly over time, making it difficult to detect changes in the system recovery time. Instead, we use the integrated grounding line flux, which shows much more variability and whose change is directly related to the MISI mechanism. As with other studies of this type, the model output is processed prior to the calculation of EWIs. This consists of aggregating the output (i.e. data binning) to remove variability with a frequency higher than that directly relevant to the internal ice dynamics considered here and thus not related to the system recovery time, together with detrending to remove nonstationarities (detrending is included in the DFA algorithm and therefore not required before calculation of the DFA indicator). Detrending was done using a Gaussian kernel smoothing function that has been shown to perform better than linear detrending (Lenton *et al.* 2012a). A smoothing bandwidth was selected that removed long term trends without overfitting the model time series. Indicators are calculated over a moving window with a length of 300 years. The optimal window length is further discussed in Sect. 4.3.

From the processed time series, we calculate three different EWIs:

1. Critical slowing down is measurable as an increase in the state variable auto-correlation. We measure this here using the lag-1 auto-correlation function (Dakos *et al.*, 2008; Scheffer *et al.*, 2009; Held and Kleinen, 2004) applied to the grounding line flux over a 300 year moving window preceding each tipping point (ACF indicator).

2. Similarly, DFA (Peng *et al.*, 1994) measures increasing auto-correlation in a time series and we apply this with the same moving window approach.

3. An additional consequence of critical slowing down is that variance will increase as a tipping point is approached (Scheffer *et al.*, 2009). We calculate variance of grounding line flux for each moving window and this can be used in conjunction with other indicators to increase robustness.

As described in Sect. 2, recovery time should tend to infinity as a tipping point is approached. This corresponds to the ACF and scaled DFA indicators reaching a critical value of one. In practice, for a complex model there are a wide variety of reasons why a tipping point might be crossed before the EWI reaches a critical value. For example, this can be a result of variability in the control variable pushing the system over a tipping point despite its long-term mean still being some distance from its critical value. For this reason, most studies adopt an alternative approach of looking for a consistent increase in the EWIs in the run up to a tipping event. This is often measured by calculating the nonparametric Kendall's $\tau$ coefficient, a measure of the ranking/ordering of a variable, which equals one if the indicator is monotonically increasing with time (Dakos *et al.* 2008; Kendall, 1948). This single value enables a simple interpretation of our results, since $0 < \tau \leq 1$ means the EWI is tending to increase with time, suggesting an imminent tipping point. The closer to one the calculated $\tau$ coefficient is, the greater the tendency for an indicator to be increasing with time and conversely a $\tau$ coefficient close to zero suggests no clear tendency for an indicator to be changing with time. We present our results in terms of both of the aforementioned criteria; whether an EWI reaches a critical value preceding the tipping point and whether the EWI is consistently increasing for a period of time before the tipping point.

## 4. Results

The quasi-equilibrium simulation shows three potential tipping points with respect to the applied melt (Fig. 4). Upon crossing each threshold, indicated by the numbered blue dots in Fig. 4, PIG undergoes periods of not only rapid but (as we show below) also self-sustained and irreversible mass loss. At this stage, relying only on a record of changes in ice volume resulting from an increasing forcing (solid black line in Fig. 4), one can only speculate that these are indeed tipping points and more analysis is necessary to confirm this hypothesis, as we go on to later. The last of the three events causes a permanently irreversible

collapse within the entire model domain (Fig 4a). We focus our results on these three major changes in the glacier configuration and ignore any possible smaller tipping points that do not result in significant grounding line retreat or changes in ice volume. We increase basal melt rates gradually and in a quasi-steady-state manner to ensure that successive retreat events can be isolated, and their effects do not overlap during the simulation. A more rapidly increasing forcing could lead to one tipping point cascading into the next and result in three individual tipping points being misinterpreted as only one event.

Grounding line positions before each of these retreat events and after the final collapse are shown in Fig. 3. Events 1 and 2 each contribute approximately 20mm of sea-level rise while event 3, which arises after slightly more than doubling current melt rates, contributes approximately 100mm. The actual sea level rise that would result from this third and largest event is

likely to be larger since in our simulation the effects stop at the domain boundary and in reality neighbouring drainage basins would be affected.

## 4.1 Early warning for the marine ice sheet instability

The three periods of MISI-driven retreat after a tipping point has been crossed can be identified clearly using EWIs (Fig. 5). The ACF indicator increases and tends to one as the tipping points are approached (Fig. 5a-c), indicating a tendency to an infinitely long recovery time as predicted by theory. We calculate Kendall's $\tau$ coefficient to identify trends in the indicator, with a value of one representing a monotonic increase in the indicator with time. The positive Kendall's $\tau$ coefficient shows that in all three cases, the lag-1 auto-correlation increases before the onset of unstable retreat. Furthermore, the ACF indicator reaches a critical value of one relatively close in time to when the MISI event begins.

These findings are supported by the DFA indicator, described in Sect. 2. The Kendall's $\tau$ coefficient indicates a significant increase of the indicator when approaching the tipping points and the indicator trends towards a critical value of one. We show the change in normalised variance calculated over each time window and in all cases this increases ahead of the tipping points being crossed with a positive Kendall's $\tau$ coefficient. The increase in variance gives greater confidence to the findings of the other two EWIs, although variance cannot be used directly to predict when that threshold will be crossed since it does not approach a critical value before a tipping point is crossed.

## 4.2 Hysteresis of Pine Island Glacier

In order to verify that we have correctly identified tipping points using the EWIs, we run the model to steady state for a given melt rate to search for hysteresis loops that indicate the presence of unstable grounding line positions. These simulations start from either the initial model setup (advance steady state) or the configuration just prior to the final tipping point (retreat steady state). The model is run for a range of melt rates between these two states, with the mean melt factor held constant and the same natural variability applied as in the forward simulation, until the modelled ice volume reaches a steady state. The first two tipping events show relatively small but clearly identifiable hysteresis loops (Fig. 4b), for which recovery of the grounding line position requires reversing the forcing beyond the point at which retreat was triggered (i.e. as shown in Fig. 1c). The third event marks the onset of an almost complete collapse of PIG (Fig. 4a). Unlike the previous two, this collapse cannot be reversed to regrow the glacier for any value of the control parameter. This is an example of a permanently irreversible tipping point, as shown in Fig. 1d. Note that this permanent irreversibility is only true for the glacier modelled in isolation and by expanding the domain it would presumably be possible for other catchments that may not have collapsed to enable this glacier to regrow.

## 4.3 Robustness of the indicators

We carry out several tests to assess the robustness of the EWIs and their sensitivity to the processing that is done on the model output prior to calculating each indicator. Two parameters in this processing step are the bin size into which data are aggregated

and the bandwidth of the smoothing kernel that removes long term trends in the time series. To check that the increasing trends in our indicators are a robust feature of our results, regardless of these choices, we conducted a sensitivity analysis. The parameters were varied by +/- 50% and the indicators were recalculated for each resulting time series. As before, we assess the utility of an indicator by whether it shows an increasing trend before each tipping point, as measured by a positive Kendall's $\tau$ coefficient. The results of this sensitivity analysis are presented for each MISI event in Fig. 6. Kendall's $\tau$ coefficient is positive for all tested combinations of parameters and all MISI events, although MISI event 2 is particularly insensitive to these parameter choices whereas the spread in Kendall's $\tau$ coefficient is greater for the other two events.

In general, critical slowing down will only occur close to a tipping point. Determining how close to a tipping point a system must be in order to anticipate the approaching critical transition, i.e. the prediction radius, is an important question and also informs the selection of palaeo-records that could be used to detect an upcoming MISI event. We show results for a window size of 300 years (i.e. a record length of 600 years), which is the shortest window size for which the DFA indicator provides a clear prediction for all tipping events. We explored the prediction radius of our model by calculating Kendall's $\tau$ for the ACF and DFA indicators and the variance for a range of window lengths, see Fig. 7. For the main tipping event, preceded by the longest stable period, the indicators gradually lose their ability to anticipate a tipping event as more data is included further from the event. The same is true for the two smaller tipping events, but the drop off is quicker such that the indicators break down for window lengths > 500 years. These results suggest that the prediction radius is relatively small and window sizes that are too large, and hence include data far from a tipping point, become less useful for the application of EWIs.

In addition to a sensitivity analysis, it is important to check that trends in the calculated indicators are statistically significant and not the result of random fluctuations. We follow the method originally proposed by Dakos *et al.* (2012) and produce surrogate datasets from the model time series that have many of the same properties but should not contain any critical slowing down trends. We generate 1000 of these datasets using an autoregressive AR(1) process based surrogate. For each of these datasets we calculate the ACF and DFA indicators and variance in the same way as with the model time series and then estimate the trend with values of Kendall's $\tau$ coefficient. We calculate the probability of our results being a result of chance for each indicator and for all three combined as the proportion of cases for which the surrogate dataset was found to have a higher correlation than the model time series. We find that $P<0.1$ in all but one instance for the ACF and DFA indicators but variance trends were generally less significant (Table 1). However, the combined probability that all three indicators would be equally positive as a result of chance was less than 0.02 for the first MISI event and less than 0.005 for the second two events.

**5. Discussion**

The indicators we have tested provide early warning of tipping points as they are approached in our transient simulation with gradually increasing melt rates. Tipping points driven by the MISI represent potential 'high impact' shifts in the earth climate

system, since they may lead to considerable changes in the configuration of the Antarctic Ice Sheet that are effectively irreversible on human timescales. Computational models are frequently used to forecast future changes of the Antarctic Ice Sheet in response to various greenhouse gas emission/warming scenarios. Predictive studies of this kind sometimes label periods of rapid retreat as 'unstable' without further analysis of the type done here (e.g. Joughin *et al*. 2014; Ritz *et al.* 2015; Favier *et al.* 2014) or avoid making this diagnosis altogether (DeConto and Pollard, 2016). Here, we have demonstrated that EWIs robustly approach critical thresholds preceding tipping points driven by the MISI. Our results show that EWIs can be used as a method to identify instabilities without the need of the aforementioned modelling approach based on computationally expensive equilibrium simulations.

It is important to clearly understand what critical threshold is identified by the EWIs. In Fig. 4 the simulated steady-states show the crossing of the tipping point earlier than identified by the indicators in the transient simulation. Since the time-scales of ice flow are longer than the forcing time-scale, the ice-sheet system modelled here does not evolve along the steady-state branch (as shown schematically in Fig. 1c). Relaxation to a steady-state takes centuries to millennia in the simulations. This means that while technically the critical value of the control parameter (basal melt rate) might have already been crossed, the glacier state could still be reverted in the transient simulation at that point, if the basal melt rate was reduced below the critical threshold. This is true until the system state variable crosses its critical value (point $x_t$ in Fig. 1c) – and this is the point identified by the EWIs. This complication in interpreting EWIs is inherent to ice dynamics because of its long response time scales. We find that both the ACF and DFA indicators not only increase as a tipping point is approached, as shown by positive Kendall coefficients, but also generally approach the critical value of one, although with varying degrees of precision (Fig. 5). This enhances their predictive power, since by extending a positive trend line it is possible to approximate what value of the control parameter will eventually cause a tipping point to be crossed. While our experiments in Appendix A showed that critical slowing down can accurately predict onset of tipping points in an idealised setup, applying this method to a more complex case study may have failed and in this context our finding that these indicators largely retain their predictive power is very encouraging. One area of additional complexity in our model of PIG compared to the setup in Appendix A is the bed geometry, which is obtained from observations and so much less smooth than the synthetic retrograde bed used in the MISMIP experiments. We explored how the addition of 'bumpiness' to bed geometry affects the performance of EWIs and found that this reduces how clearly we can resolve the change in response time (Appendix A). This effect may account for the fact that EWIs do not precisely reach a value of one at the bifurcation point but confirming this would require further testing.

There are several important caveats to the use of EWIs as presented here. Firstly, and as explained above, the tipping point identified is that of the transient system not in steady state. Although the transient behaviour is arguably of greater societal relevance and an ice sheet is unlikely to ever truly be in steady state, this is an important distinction to make. Secondly, the predictive power of this method decreases as the distance to tipping increases and must eventually break down altogether. This effect can be clearly seen in Fig 7 as the Kendall's $\tau$ coefficient decreases with increasing window length. Thirdly, there is a

risk of so-called 'false alarms' and 'missed alarms' (Lenton, 2011). False alarms, whereby a positive trend in an indicator that is incorrectly interpreted as a tipping point being imminent, can occur for a wide variety of reasons. First and foremost, interpreting EWIs requires robust statistical analysis and judicious data processing to ensure that the response time being

measured is that of the critical mode (Lenton, 2011). It is possible that rising autocorrelation is a result of other processes, and using more than one indicator together with changes in variance can help mitigate this risk (Ditlevesen, 2010). It is also possible for a tipping point to be crossed with no apparent warning i.e. 'missed alarms'. This could happen if the internal variability in a system is high so that it changes state before a bifurcation point is reached, or similarly if the forcing is too sudden. This last point is particularly pertinent, since we intentionally perturb our model slowly and do not explore how a change in forcing rate

affects the performance of our chosen EWIs. Increasing the forcing rate might present further difficulties in identifying tipping points by leading to multiple tipping points being crossed coincidentally. Since in this methodology the control parameter is not held constant after a tipping point is crossed, then if that parameter changes sufficiently during the time it takes for one tipping event to conclude it might reach a second threshold while the first event is still underway, disguising the fact that two distinct tipping points had been crossed. Changing the control parameter very slowly alleviates this issue, since it will only

have altered slightly during the time it takes for a tipping event to happen. This issue, along with the related issue of cascading tipping points, is one that we try to avoid in our experiments to simplify the analysis but is known to influence EWI performance (Dakos *et al.*, 2015, Brock and Carpenter, 2010). Despite our use of a very slow forcing rate, it is possible that more than three tipping points exist in our model configuration that we did not identify. Finding all possible tipping points would necessitate infinitesimally small changes in the control parameter, either in the steady-state or transient simulations,

greatly increasing computational cost but with little benefit in terms of detecting tipping events that constitute substantial mass loss.

In this paper we have presented an application of EWIs on model output to anticipate tipping points. This is a useful approach in and of itself, since it could be used in model studies to detect bifurcations in the system with minimal computational expense,

or to check whether a model might be on a trajectory to cross a tipping point at some point in time beyond the simulation. Alternatively, it may be possible to use this method on observational data, palaeo records, or some combination thereof. This raises the question of what data might qualify as useful for the application of EWIs, which can be broken down further into (1) the type of data needed and (2) the length of record necessary. As mentioned previously, ice volume or related measures of an ice sheet's size do not show sufficient variability for information on the recovery time to be extracted. Ice speed however

can change significantly over very short timescales, for example many ice streams show large variability over timescales as short as tidal periods (Anandakrishnan *et al.,* 2003, Gudmundsson, 2006, Minchew *et al.,* 2017). Ice flux was chosen in this study since it is closely related to the MISI mechanism and because flux is proportional to velocity, but it is possible that other metrics related to ice velocity might also exhibit critical slowing down in a similar way. With regards to record length, we find in this study that early warning of tipping points becomes less reliable (with low or even negative Kendall's $\tau$ coefficient) for

a moving window size shorter than 200-300 years. However, this does not mean that this represents the minimum window size

in general and is likely sensitive to a number of the choices in our methodology. For example, this value is likely to be sensitive to the rate of forcing applied to the system. In the limiting case of a forcing rate approaching zero, the necessary window length must increase since EWIs are only expected to work relatively close to the tipping point. Both of these points require further study in order to establish suitable datasets for prediction of MISI onset.

## 6. Conclusions

Conducting quasi-steady numerical experiments, whereby the underside of the PIG ice shelf is forced with a slowly increasing ocean-induced melt, we have established the existence of at least three distinct tipping points. Crossing each tipping point initiates periods of irreversible and self-sustained retreat of the grounding line (MISI) with significant contributions to global sea level rise. The tipping points are identified through *critical slowing down,* a general behavioural characteristic of non-linear systems as they approach a tipping point. EWIs have been successfully applied to detect critical slowing down in other complex systems. We here show that they robustly detect the onset of the marine ice sheet instability in the simulations of the realistic PIG configuration which is promising for application of early warning to further cryospheric systems and beyond. While the possibility of PIG undergoing unstable retreat has been raised and discussed previously, this is to our knowledge the first time the stability regime of PIG has been mapped out in this fashion. The first and second tipping events are relatively small and could be missed without careful analysis of model results but nevertheless are important in that they lead to considerable sea level rise and would require a large reversal in ocean conditions to recover from. The third and final tipping point is crossed with an increase in sub-shelf melt rates equivalent to a +1.2°C change in ocean temperatures and leads to a complete collapse of PIG. Long-term warming and shoaling trends in Circumpolar Deep Water (Holland *et al.*, 2019), in combination with changing wind patterns in the Amundsen Sea (Turner *et al.*, 2017), can expose the PIG Ice Shelf to warmer waters for longer periods of time, and make temperature changes of this magnitude increasingly likely.

## Appendix A: Flowline experiments

The MISI has been a major focus of modelling efforts within the glaciological community in recent years. In an effort to assess how ice-flow models capture this behaviour, a model inter-comparison experiment was performed to calculate the hysteresis loop of advance and retreat of a marine ice sheet on a retrograde slope, known as MISMIP experiment 3 (referred to as EXP 3 hereafter, Pattyn *et al.*, 2012). As a first step to establishing whether critical slowing down can be observed prior to the MISI, we undertook a slightly modified version of this experiment using the Úa ice-flow model (Gudmundsson 2012, Gudmundsson 2013, see methods). In our modified experiment, the marine ice sheet is forced towards tipping points through step perturbations in the control parameter as before, but with smaller steps and the additional constraint that the model must be in steady state after each perturbation before moving onto the next. In this experiment the chosen control parameter is the ice rate factor, a parameter linked to ice viscosity and temperature.

Following each perturbation in the ice rate factor, we analyse the e-folding relaxation time ($T_R$) of the state variable (in this case, grounding line position) to directly extract the recovery time of the model as it approaches each tipping point (both advance and retreat). Theory predicts that $T_R \to \infty$ close to a tipping point and that the point at which $T_R^{-2}$ (as plotted versus the control parameter) reaches 0 thus identifies the critical value of the control parameter, beyond which a tipping point is crossed (Wissel 1984). We show this plot for both the advance and retreat scenarios of EXP 3 in Fig. A1. In both cases the relaxation time decreases as predicted by theory, even far from the tipping point. A linear fit through the last six perturbations yields a good agreement with theory and accurately predicts the critical value of the control parameter when compared to the analytical solution (red arrows in Fig. A1) given by Schoof (2007). Critical slowing down still occurs outside of this range (equivalent to a change in ice temperature of >5 °C) but using these more distant points to forecast the tipping point would yield a less accurate prediction. These results therefore provide some insight into how far from the tipping point we can expect the predicted linear response.

One major simplification in this idealised experiment is the bed geometry, which is synthetic and arguably unrealistically smooth. To test whether the addition of 'bumpiness' to the bed affects how accurately the critical value of the control parameter can be predicted, we conducted further experiments in which the bed was made successively less smooth. One simple but flexible way to generate the desired roughness is to add Perlin noise to the bed. Perlin noise is a commonly used method in terrain generation that adds noise at a number of levels with successively smaller wavelengths and amplitudes. The number of levels is denoted by the octave, the rate at which each octave changes frequency is the lacunarity and the rate at which each octave changes amplitude is the persistence. We made the common choice of a lacunarity greater than one and a persistence less than one, meaning that each octave adds noise of higher frequency and lower amplitude. For a starting octave amplitude of 25m the difference between the analytical solution and linear fit is less than 1%, but this grows to ~5% with an amplitude of 50m (note the change in height from peak to trough in the retrograde region of the smooth bed is ~120m). This suggests that more realistic bed geometries with increased roughness might make the task of predicting tipping points more challenging than it is in this simplified case.

**Code availability**

The source code of the Úa ice-flow model is available from https://github.com/ghilmarg/UaSource (last access: 30 June 2020) and raw model output is available from the authors upon request.

## Author contribution

SHRR and RR conceived the study, SHRR conducted the modelling experiments, JFD contributed to the statistical analysis and surrogate time series, JDR provided an initial model setup. SHRR and RR wrote the manuscript with contributions from all authors.

## Competing interests

The authors declare that they have no conflict of interest.

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

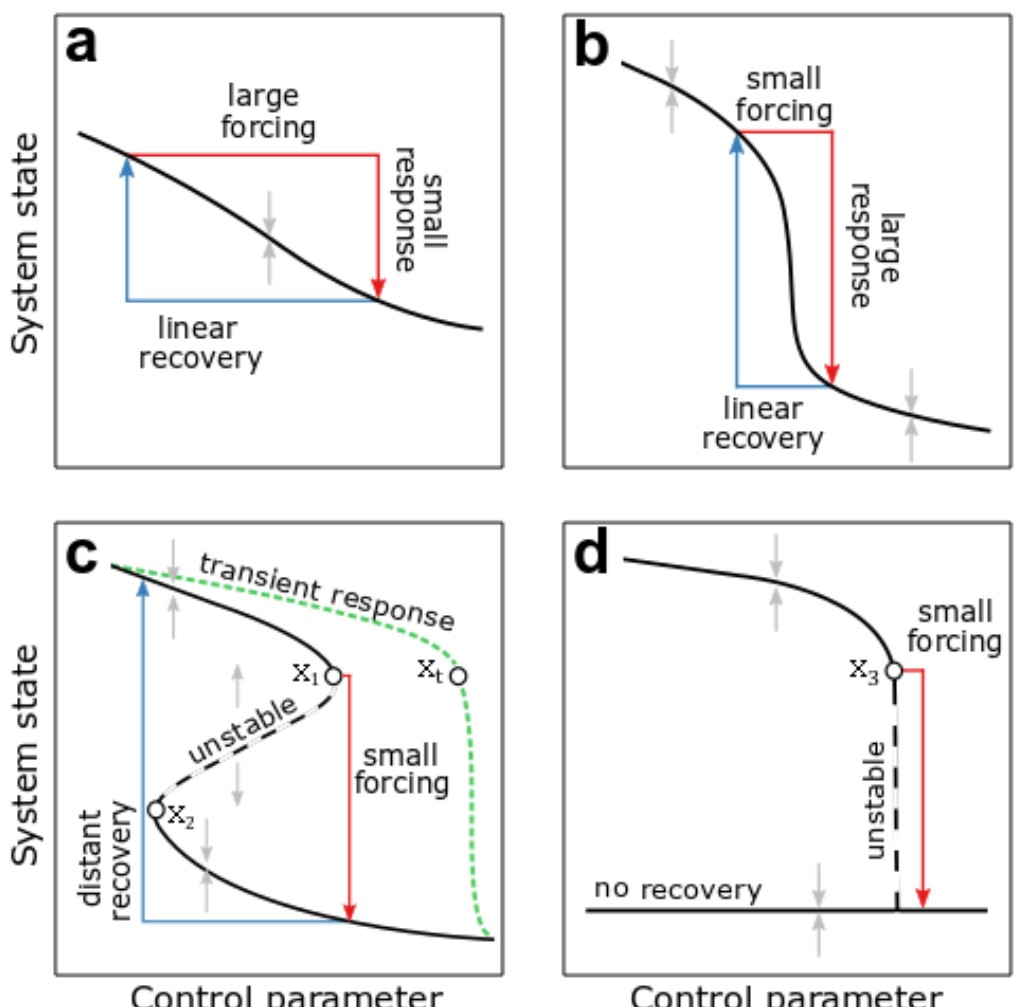

Figure 1. Possible range of behaviours for a system state (e.g. ice flux) in response to perturbations in a control parameter (e.g. ocean temperature). A system can respond to a perturbation (a) in a linear way that is directly recoverable with a reversal of the forcing, (b) with a large response to a small perturbation but that is still directly recoverable, (c) with a large response to a small perturbation that is irreversible (hysteresis behaviour), and (d) with a large response that is irreversible for any change in the tested range of the control parameter, a behaviour we refer to as permanently irreversible (no recovery possible even if forcing is reversed well below the initial level). Tipping points are crossed only in panels (c) and (d) and are indicated by $x_1$, $x_2$ and $x_3$. Panel (c) also shows a transient response in which the system state lags behind changes in the control parameter as is the case for ice sheets and thus crosses the irreversible system state at a later point, $x_t$.

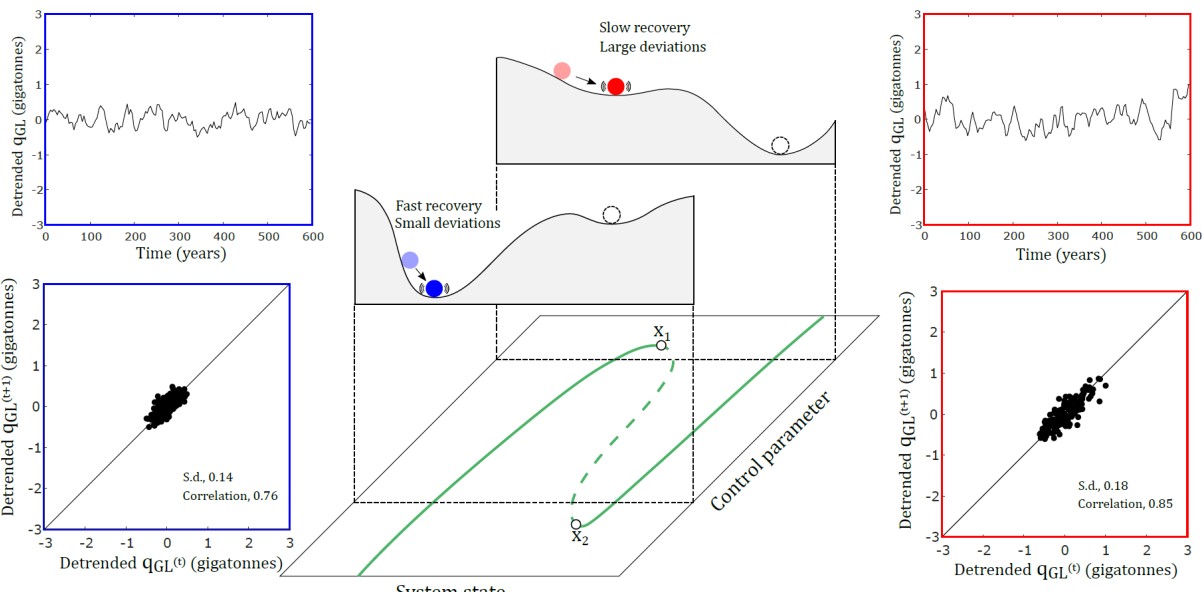

**Figure 2. Critical slowing down can serve as an indicator that the system is approaching a tipping point. This can be understood conceptually using the common 'ball on a slope' analogy (middle panel), where the ball represents the system state and minima are stable equilibrium states. Two example cases are superimposed onto their corresponding positions in the hysteresis plot of MISI shown by the green line and equivalent to Fig. 1c. The processed model results demonstrate how critical slowing down manifests itself, as shown in the blue and red panels at the sides. If the system is far from a tipping point (blue case), the system state (which is in this study the grounding line flux $q_{GL}$, upper left panel) recovers quickly from perturbations in the control parameter (which is here the basal melt variability). This means that from one measurement (at time t) to the next (at time t+1) the grounding line flux changes rapidly and has a low lag-1 auto-correlation (lower left panel). Conversely, close to a tipping point (red case), critical slowing down manifests and the system state responds more slowly to perturbations in the control parameter (upper right panel). Since the state variable is changing more slowly, successive measurement are more similar, resulting in a higher lag-1 auto-correlation (lower right panel).**

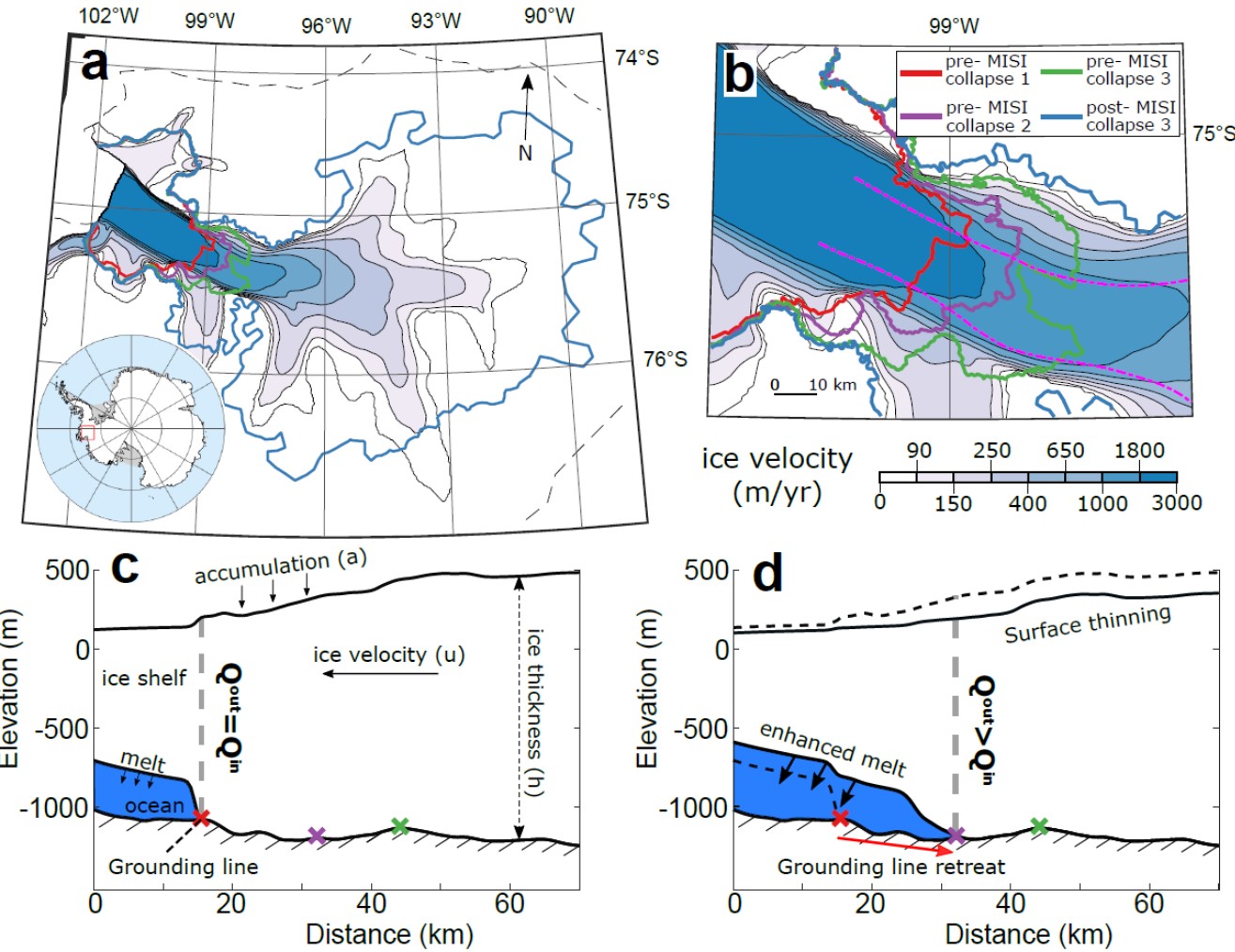

**Figure 3. Marine Ice Sheet Instability events for Pine Island Glacier.** Shown are (a) grounding line positions before and after the three MISI driven glacier collapses with (b) a zoom to the initial events (coloured lines). The colormap indicates initially modelled ice velocity and the model domain boundary is indicated by a dashed black contour in panel a . Panels (c) and (d) show a transect through the main trunk of PIG, calculated as an average of properties between the two dashed magenta lines in (b). The vertical

section along the transect is shown (c) at the initial steady state where fluxes ($Q_{in}$ and $Q_{out}$) are in balance and (d) during a MISI event where retreat causes an increase in $Q_{out}$, pushing the glacier to be out of balance and leading to further retreat.

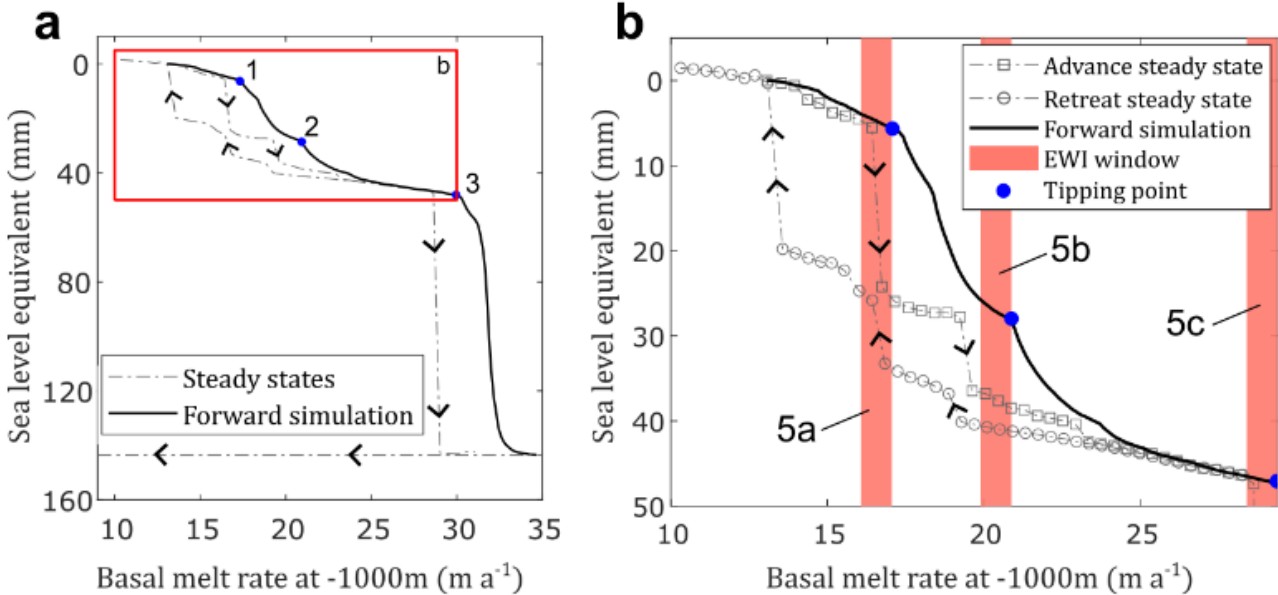

**Figure 4. Change in system state in terms of sea level equivalent ice volume as a function of the control parameter, which is the melt rate at the ice-ocean interface. (a) The model is run forward with a slowly increasing basal melt rate (solid black line) and shows three distinct tipping points (blue dot). From the start of the transient simulation to the third tipping point is approximately 10kyrs. The steady states for a given melt rate in both an advance and retreat configuration are plotted as dashed grey lines, arrows indicate the direction of the hysteresis. Panel (b) focuses on the model response before the larger tipping point (event 3) and shows the three windows that we analyze for early warning indicators as shaded red boxes (Fig. 5). Circle and square symbols represent steady state configurations for a given forcing and the dashed grey line is a linear interpolation between these points. Each step in melt rate for the steady state runs from ~10 to ~30 m a$^{-1}$ is approximately equivalent to 0.4m/yr of basal melting, or 250 years in the transient simulation. The lower branch in panel (a) represents a simulation starting from the PIG configuration after the third major retreat event and reverses the basal melt rate factor to its lowest value, showing no recovery in ice volume.**

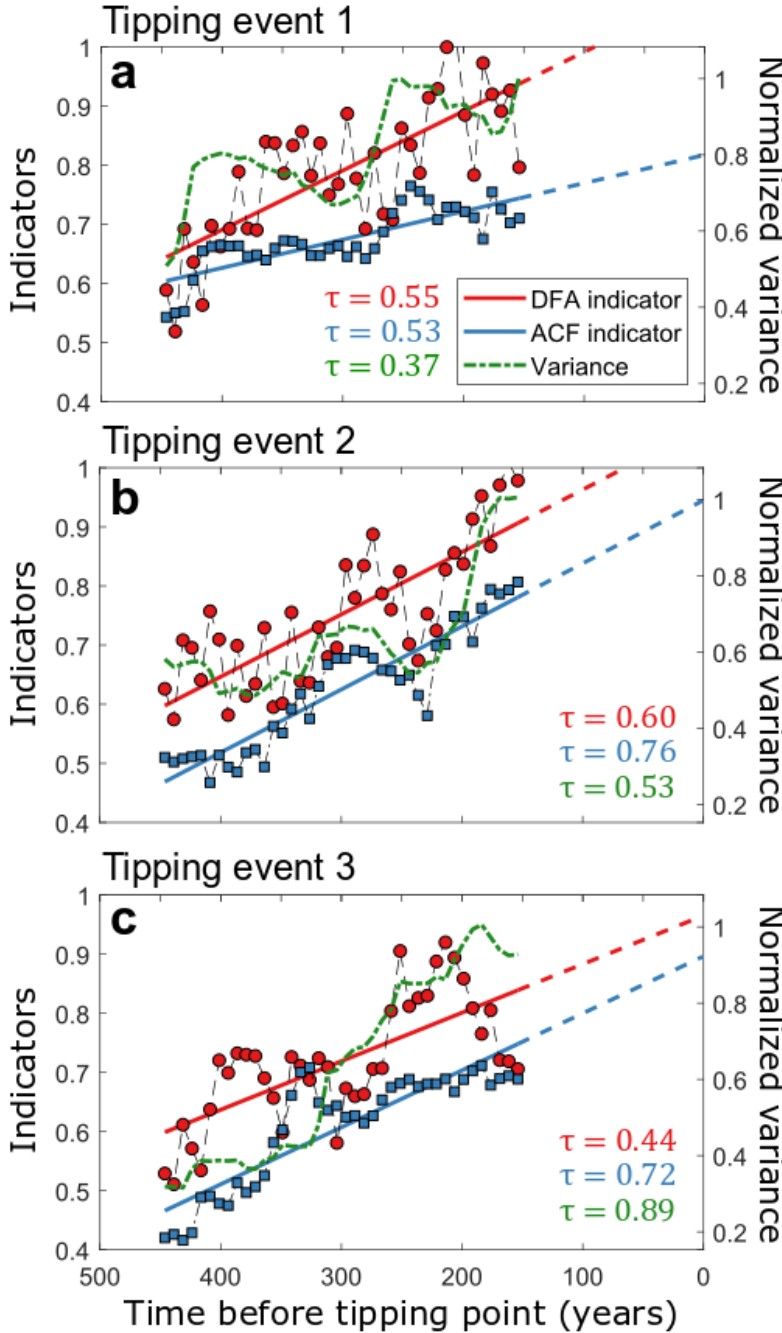

**Figure 5. EWIs for the marine ice sheet instability in Pine Island Glacier. Each panel shows the EWIs preceding each of the three MISI tipping event marked in Fig 3b, along with the linear trend extrapolated to the point in the simulation when the respective tipping event occurs. Increasing trends in all indicators are shown by a positive Kendall's $\tau$ coefficient which measures the correlation between each indicator and time between -1 and 1.**

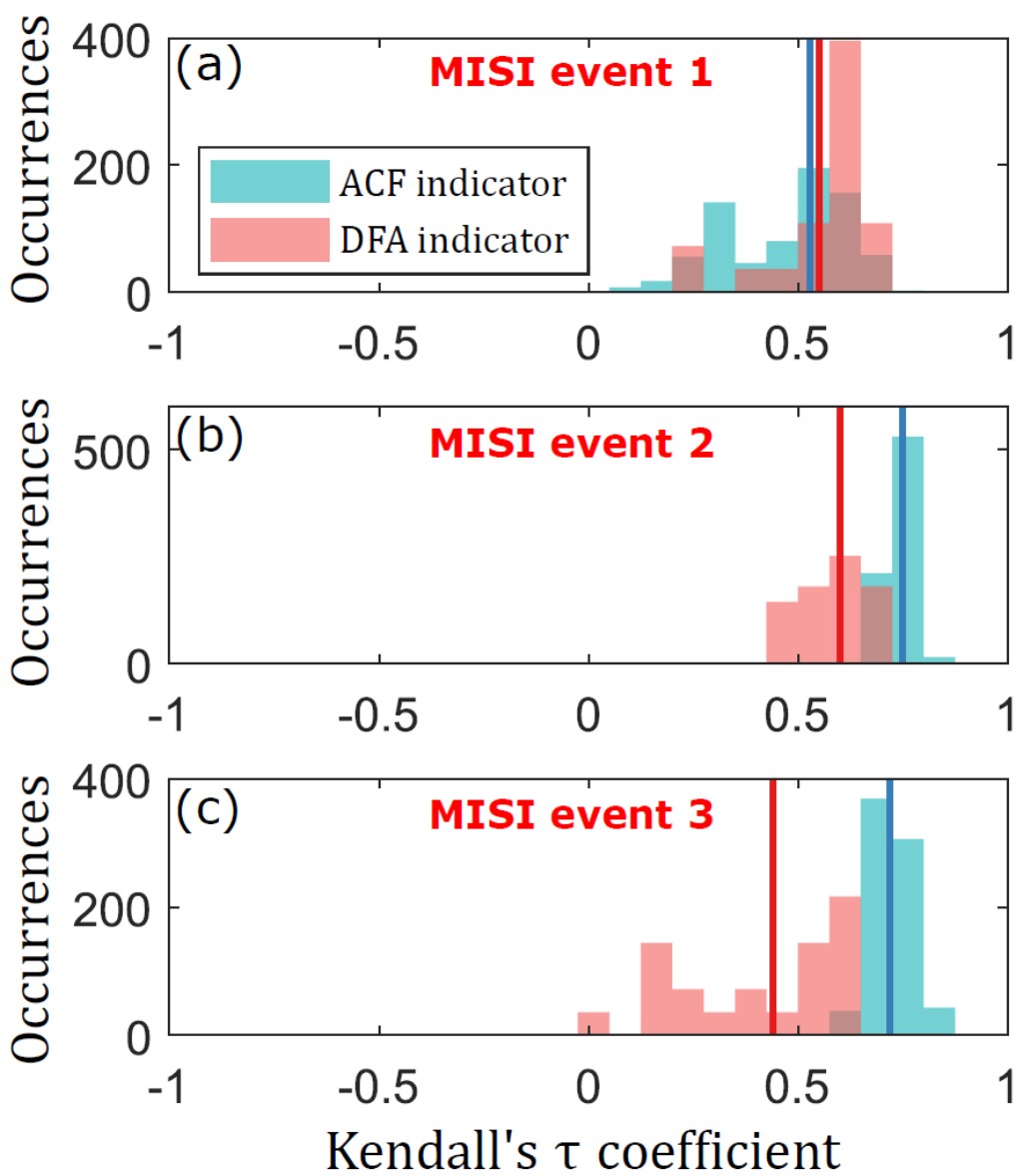

**Figure 6. Sensitivity analysis for the ACF and DFA indicators. Each occurrence is the Kendall's τ coefficient for a different choice of filtering bandwidth and data aggregation. The solid red and blue lines show the Kendall's τ coefficient for the DFA and ACF indicators respectively, as calculated for the choice of parameters used in Fig. 5.**

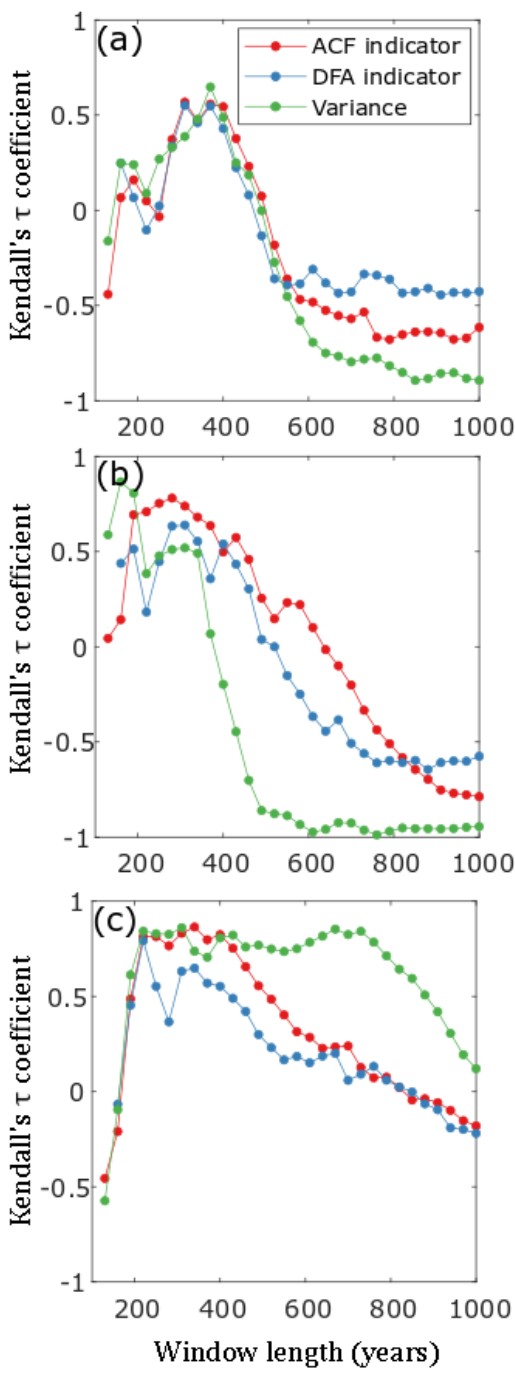

**Figure 7. The effect of window length on the predictive power of EWIs for the MISI. The three panels show the change in Kendall's τ coefficient as calculated for each indicator versus window length for MISI events 1, 2 and 3 (panels a, b and c respectively).**

| Event Number | Indicator name | Indicator value | Probability | Total Probability |
|---|---|---|---|---|
| **MISI event 1** | DFA | 0.55 | 0.041 | 0.0198 |
| | ACF | 0.53 | 0.122 | |
| | Variance | 0.37 | 0.315 | |
| **MISI event 2** | DFA | 0.60 | 0.022 | 0.0030 |
| | ACF | 0.76 | 0.012 | |
| | Variance | 0.53 | 0.207 | |
| **MISI event 3** | DFA | 0.44 | 0.099 | 0.0044 |
| | ACF | 0.72 | 0.026 | |
| | Variance | 0.89 | 0.018 | |

**Table 1. Probability of the Kendall's $\tau$ correlation for each indicator being a result of chance. One thousand surrogate time series of the state variable are generated and the indicators and Kendall's $\tau$ correlations calculated for each one. The probability of a Kendall's $\tau$ value is then the fraction of these surrogate time series with a higher correlation coefficient. The total probability is the fraction of surrogates for which all three indicators have a higher correlation coefficient than is observed in the original model time series.**

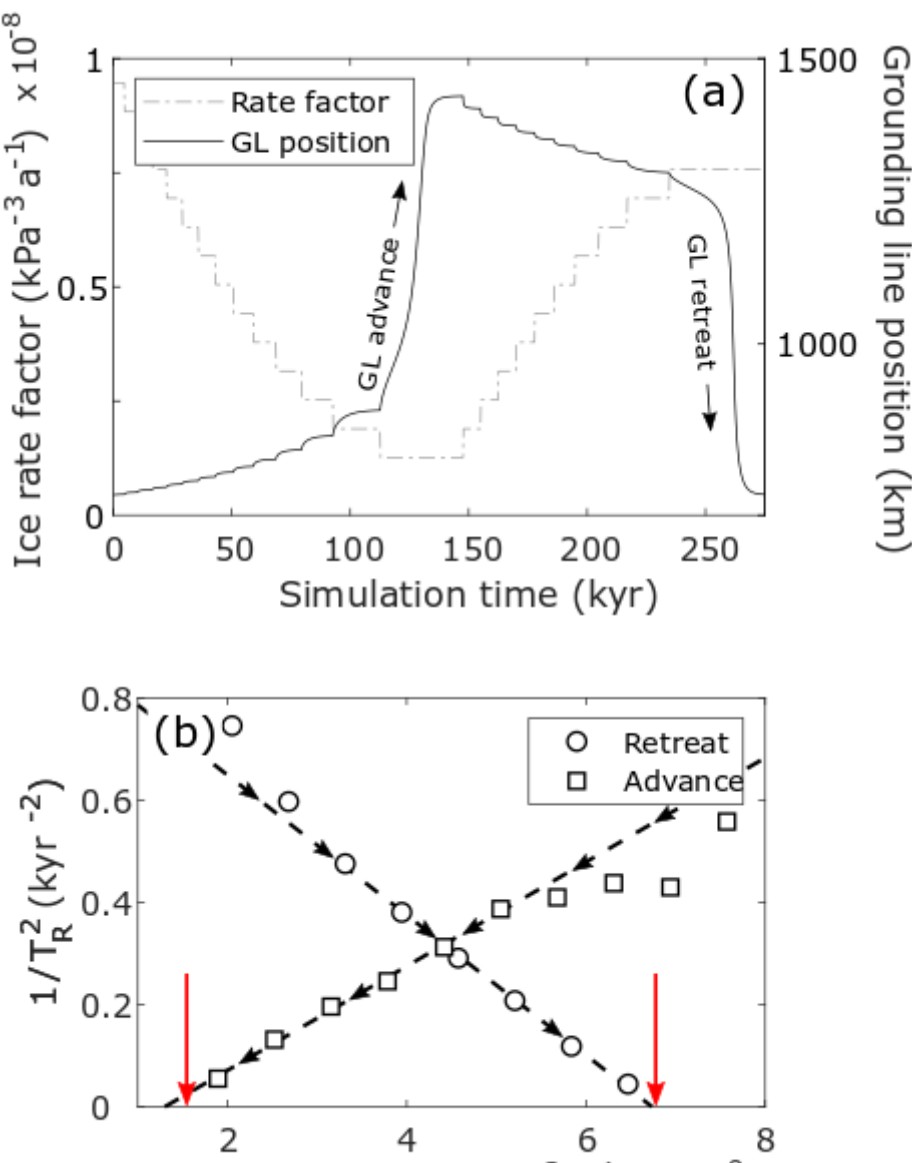

**Figure A1. Results of EXP 3, showing change in GL position with time resulting from step perturbations in the ice rate factor (panel a). The calculated inverse relaxation time for each corresponding step change in rate factor in both the advance (square symbols) and retreat (circular symbols) phase is shown in panel b. The dashed line in panel b is a line of best fit, calculated for the five steps in rate factor that preceded the advance or retreat MISI phase. Red arrows indicate the rate factors for which the analytical solution**
**predicts a MISI event and black arrows show the direction of the forcing towards each tipping point.**