# Peer review of "The tipping points and early-warning indicators for Pine Island Glacier, West Antarctica"

_The Cryosphere, 2020_

## Referee Comment (RC1) · Alexander Robel (Referee) · 9 Sep 2020

This manuscript explores the stability of Pine Island Glacier under forcing from ocean melt, using a high order model of marine ice sheet dynamics. Additionally, this is the first study to bring the concept of Early Warning Signals (EWS) to the stability transition known as the "marine ice sheet instability" (MISI). Though EWS have been explored in a limited degree in some other areas of glaciology, this is an interesting and useful application to the MISI problem, which has recently been a focus of intense study in the glaciology community.

[Figure]

The central concept and design of this study are sufficiently novel and important that it eventually should be published in The Cryosphere, though I think it requires some revision first. In particular, since this is the first application of EWS to the MISI problem, it needs to be clear why exactly EWS are a useful tool for studying marine ice sheet stability. Additionally, the methodological details of EWS, while established in the dynamical systems literature, are not well known in glaciology. If the authors wish other glaciologists to follow their lead in using this approach (which I think should be the case), then they need to do a better job explaining the methods they use and the assumptions inherent to these methods. I lay out these critiques in more detail below through major and minor suggestions:

Major:

1. Here is the question that you should answer in this manuscript: Why/When are EWS a useful tool for understanding MISI at a particular glacier? At the moment, my interpretation (perhaps erroneous) of the purpose of EWS laid out in this manuscript is to show that there is a bifurcation (in fact several) in the model. However, you don't need EWS to show that this is the case when you have a model available, since you have the quasi-steady simulations which show the bifurcation structure of PIG. Rather, the point of EWS is to detect a bifurcation before it occurs. You do so in the context of your model, however, solely within the context of a predictive model, EWS are not strictly necessary, because the model can be run forward to determine whether a bifurcation will occur with continued forcing along some trajectory (this is the point of physical models!). However, what you could argue here is that your study is a proof-of-concept to indicate the circumstances under which we would expect to detect EWS in observations, which would be immensely useful for the community. This is what I find currently

lacking in the study - any discussion of the implications of your study for observations. For example, you touch on this issue later in the paper about the fact that in the real world, ice sheets do not stay on the stable manifold because forcing is much faster than the response time, but then don't really explore whether this makes EWS useless in practice (I think no, but that isn't my take away from the current way its written). Another issue (which you don't remark on) is the fact that a 300 year averaging window for the EWS indicators is not super useful when the entirety of the observational record is 40 years long (perhaps a bit longer if we include some lower quality historical obs). This is all to say that showing EWS exist in a model is not very insightful in of itself if it doesn't provide some indication for what we should be looking for in observations (since again, we already know that there are bifurcations associated with MISI in models).

2. You haven't necessarily explained why EWS show up in certain types of systems. To me, this is key to then explaining why you are calculating these things (ACF, variance, etc). In a canonical saddle-node bifurcation, we expect the stable eigenvalue of the linearized system state to smoothly decrease towards zero as you approach the bifurcation, which causes weaker damping of noisy forcing back towards the stable manifold. So, do EWS only occur where there is a saddle-node bifurcation? How do we know that MISI at PIG has such a smooth approach to the bifurcation? i.e. if the eigenvalue associated with the stable mode is controlled at first order by bed topography (which it is in the canonical formulation of MISI, see Schoof 2012 and others), then does the topography in the vicinity of the bifurcation need to vary smoothly towards a bedrock peak to produce EWS? Maybe these are questions for another study, but there needs to be some indication that you have grappled with the question of why you expect EWS to occur for PIG.

3. Related to the issue above, more of the detail about how the EWS indicators are calculated would be helpful to bring into the main text, since this is a topic most

TC readers are not familiar with. How do you ensure statistical significance? Could we see EWS away from a bifurcation by chance? Why not?

4. You say that you force your model with variability from a "surrogate model" based on ocean variability in the Amundsen Sea region, but don't provide further details. First, more detail is needed on the surrogate model. Second, presumably this surrogate model produces ocean variability with significant power in the decadal range, as many studies (e.g. from Jenkins and others) have found that such variability is important in this region. However, the typical formulation of EWS (e.g. Lenton et al. 2008, and previous studies) assumes a martingale process for the noise forcing (i.e. white noise) which is not the case here. Can you explain why this doesn't affect your interpretations of EWS indicators? Third, it is unclear whether the steady-state and quasi-steady-state simulations used to make the bifurcation diagrams in Figure 3 include noise forcing. If not, then this is concerning, because it is well known that marine ice sheets have a different steady-state with and without noise in the forcing (e.g. Robel et al. 2018, Hoffman et al. 2019, Mikkelsen et al. 2018 (but for non-marine ice sheets)). This could be quite important in your simulations, since the location of the bifurcation is important to know for calculating EWS.

Minor:
Line 23: please define what you mean by "tipping element" for the uninitiated
Line 29: grounding line flux
Line 47: the 2012 Schoof JFM paper makes more sense to cite here
Line 48: what do you mean by catastrophic?
Line 89: So is accumulation held constant in time or does it have a seasonal cycle? Be more specific here, because its important to know if there is variability in more than just ocean melt.
Line 119: Again, to be clear, it isn't necessarily the case that EWS exist for all tipping points (if we define any bifurcation as a tipping point).

Line 126: it would help here to explain exactly how you force the simulations to produce the grey dashed and black lines in Figure 3 (also in caption). How fast is the forcing? How large are the step increments to determine the steady-states? How do you determine when there is a tipping point (just a large enough jump in the grounding line?)? What if you have a tipping point that causes the grounding line to retreat only a small amount?

Line 119-129: This whole paragraph is confusing

Line 136: What do you mean by short tem "weather noise"? Isn't this the thing that is detected by EWS? Line 184: What do you mean by "equivalent to a random walk"?

Line 197-198: is this hysteresis related to the domain extent? If you aren't simulating flow from outside the domain, then when the whole domain collapses you won't be able to regrow the glacier for any parameter (because theres no catchment).

Line 203-205: what about much shorter windows? (related to point 1 above related to observations). How short of a window would actually be calculable from observations (a decade? does this start to run into the AC time scale of forcing?)

Line 214-215: this is a confusing sentence which leads me to think that you are saying the EWS are not actually "early". To what extent does this depend on the speed of the trend in forcing? Can you test it for different trend rates?

Figure 3a: If the black line doesn't fall on top of the gray dashed line, then the black line simulation isn't really quasi-steady. Why not call it something else? Also, it is unclear how the grey dashed line is determined, and which parts are stable and unstable?

Figure 3b: please explain the different between "tipping point" and "instability onset". I can guess that the latter has to do with the region in which an unstable manifold exists (i.e. there is hysteresis), but I'm not sure readers will necessarily pick up on this without you explaining it explicitly.

Figure 4: Looking at this (and Fig. B2), it seems clear to me that the length of time ahead of the tipping point needed to detect EWS is directly correlated to the speed of

the forcing. Yet, that isn't really made clear here or in the text.
Figure B2: can you extend this to shorter windows?

Appendix A: Related to some of the issues I raised above, it may be valuable to bring the flowline simulations into the main text to demonstrate, in a very simple system where the exact location of the tipping points are known, how EWS work.

---

## Referee Comment (RC2) · Anonymous Referee #2 · 10 Sep 2020

In this manuscript the authors seek to detect the onset of marine ice sheet instability (MISI) in model simulations of Pine Island Glacier, using techniques that have previously been applied to other complex systems. The novelty of this study lies in the application of critical slowing indicators to confirm MISI events. It provides an interesting framework for evaluating vulnerabilities in ice sheets that will be of interest to the TC scientific community. However, particularly due to its novelty within glaciology, some aspects of the paper need improving to aid the clarity, and it would benefit from further exploration/discussion of the usefulness of the techniques beyond the modelling example provided here. I have outlined these below, followed by line-by-line comments.

The paper includes a nice explanation and accompanying schematics of hysteresis; however, the critical slowing description and explanation of the indicators is less intuitive. This may partly be due to the structure – Appendix A offers a useful demonstration of how critical slowing manifests in a carefully controlled simple experiment. I think it is safe to assume most TC readers will not be familiar with these concepts, and therefore I do not think this example should necessarily be tucked away in an appendix. A diagram of critical slowing in a similar vein to Figure 1 would be helpful, or some additional annotation to Figure A1. There is a disconnect between the flowline example in Appendix A, and the methods used for determining the onset of a tipping point in the main set of experiments. Could you show how critical slowing in the flowline experiment can be demonstrated with the various indicators you use in your main experiments? This would help show how these indicators are related to the the increased recovery time from a stepwise perturbation as the tipping point is approached.

My other main comment is about the usefulness of this method in detecting early warning signals in reality. Would the 300-year optimal window size apply to other catchments? What kind of observational datasets are required to implement this analysis, in a way that would act as a useful early warning system for MISI? The measurement used in this study (grounding line flux) does not exist (at least not at the quality/resolution required here) prior to the satellite era, so what is the alternative, given the 300-year window size? In your model simulations the forcing is applied gradually in order to avoid "one tipping point cascading into the next and result in three individual tipping points being misinterpreted as only one event" (L168). What are the implications of this for detecting tipping points in observations, where the system is not necessarily able to return to a quasi-steady-state with changing forcings? Do you have a sense of whether the indicators would hold up if the forcing is more rapid? Further discussion of these issues would strengthen the paper.

Other comments:

L13: "Self-amplifying retreat" this could be considered an overstatement. Self-sustaining retreat would be more accurate (and more in keeping with language further on in the manuscript).

L18: "early warning indicators robustly detect critical slowing for the marine ice sheet instability". It might be worth removing the term "critical slowing" from the abstract, and instead using a less jargon-y alternative, e.g. "robustly detect the onset of MISI"?

L31-32: "a complex range of factors can either cause or suppress the MISI" – such as? The two papers cited refer to buttressing, what about local sea level, GIA etc? Haseloff, 2018, should be Haseloff and Sergienko, 2018.

L68-70: "Our results reveal the existence of multiple smaller tipping points that when crossed could easily be mis-identified as simply periods of rapid retreat, with the irreversible and the self-sustained aspect of 70 the retreat being missed": this seems to contradict your results and conclusions. The two smaller tipping points are not irreversible, as the system can return to previous state through stronger perturbations in the opposite direction – shown by the hysteresis loops in Figure 3.

L106: Basal melt rates: it is not clear from this paragraph whether basal melt occurs under grounded ice. How do you treat partially grounded elements? This has been shown to be important in modelling grounding line retreat (e.g. Seroussi and Morlighem, 2018, doi: 10.5194/tc-12-3085-2018).

L153-159: Unlike the other paragraphs in this section, this paragraph does not contain an outcome of your decision-making process – which of the criteria will you use?

L180-181: "Furthermore, the indicator reaches a critical value relatively close in time to when the MISI event gets underway". Clarify that the critical value is 1.

L183-185: "For this early warning indicator...". I don't understand this sentence and it seems like it would be better suited (with added detail) to the methods. I thought both indicators have a critical value of 1 (section 2.2), so why does scaling the DFA help with comparison to ACF?

L188-189: "although variance cannot be used directly to predict when that threshold will be crossed" – perhaps this is obvious, but why not? Because there is no critical

value? But crossing the critical value doesn't seem like a robust way of detecting the exact onset of MISI either, considering some of the trend lines in Fig. 4 cross x=0 before they reach the critical value?

L275: What do you mean by "i.e. a record length of 600 years" – how does that relate to the window size? Is that the minimum record length required?

L258: Basin of attraction has not been defined/explained.

Figure A1: Panel A, grounding line position in km: clarify the direction of retreat.

[Figure]

---

## Author Response (AR1)

This manuscript explores the stability of Pine Island Glacier under forcing from ocean melt, using a high order model of marine ice sheet dynamics. Additionally, this is the first study to bring the concept of Early Warning Signals (EWS) to the stability transition known as the "marine ice sheet instability" (MISI). Though EWS have been explored in a limited degree in some other areas of glaciology, this is an interesting and useful application to the MISI problem, which has recently been a focus of intense study in the glaciology community.

The central concept and design of this study are sufficiently novel and important that it eventually should be published in The Cryosphere, though I think it requires some revision first. In particular, since this is the first application of EWS to the MISI problem, it needs to be clear why exactly EWS are a useful tool for studying marine ice sheet stability. Additionally, the methodological details of EWS, while established in the dynamical systems literature, are not well known in glaciology. If the authors wish other glaciologists to follow their lead in using this approach (which I think should be the case), then they need to do a better job explaining the methods they use and the assumptions inherent to these methods. I lay out these critiques in more detail below through major and minor suggestions:

*Many thanks, Alex, for the positive evaluation of our study and the helpful comments. We replied to all comments and hope that our replies and changes made to the manuscript address all issues raised. Our replies below are indicated in red, changes to the manuscript summarised in blue and references can be found at the end of the document.*

**Major:**

1. Here is the question that you should answer in this manuscript: Why/When are EWS a useful tool for understanding MISI at a particular glacier? At the moment, my interpretation (perhaps erroneous) of the purpose of EWS laid out in this manuscript is to show that there is a bifurcation (in fact several) in the model. However, you don't need EWS to show that this is the case when you have a model available, since you have the quasi-steady simulations which show the bifurcation structure of PIG. Rather, the point of EWS is to detect a bifurcation before it occurs. You do so in the context of your model, however, solely within the context of a predictive model, EWS are not strictly necessary, because the model can be run forward to determine whether a bifurcation will occur with continued forcing along some trajectory (this is the point of physical models!). However, what you could argue here is that your study is a proof-of-concept to indicate the circumstances under which we would expect to detect EWS in observations, which would be immensely useful for the community. This is what I find currently lacking in the study - any discussion of the implications of your study for observations. For example, you touch on this issue later in the paper about the fact that in the real world, ice sheets do not stay on the stable manifold because forcing is much faster than the response time, but then don't really explore whether this makes EWS useless in practice (I think no, but that isn't my take away from the current way its written). Another issue (which you don't remark on) is the fact that a 300 year averaging window for the EWS indicators is not super useful when the entirety of the observational record is 40 years long (perhaps a bit longer if we include some lower quality historical obs). This is all to say that showing EWS exist in a model is not very insightful in of itself if it doesn't provide some indication for what we should be looking for in observations (since again, we already know that there are bifurcations associated with MISI in models).

*Both reviewers state that our paper needs to do a better job of explaining the background and methodology of EWS and why our results are useful in a glaciological context. We agree that both of these things could be greatly improved on and will endeavour to do so in a revision of the paper. The aim of this paper is a 'proof of concept', demonstrating that EWS can be found for the marine ice sheet instability in realistic geometries. The theoretically-proven hysteresis of MISI (Schoof 2007) is similar to a fold (aka saddle node) bifurcation. For this class of dynamic systems the existence of EWS can be theoretically proven. However, it is by no means clear that EWS can be still detected in more complex, realistic systems, as for example discussed in Dakos et al., (2015). We show for the first time for a realistic geometry (including for example ice shelf buttressing that is not considered in the theoretical hysteresis for MISI) that indeed EWS can be detected.  We will add more discussion regarding the potential for observations of ice sheets to be used in the context of EWS since we agree that this is arguably the most important potential outcome of our findings. Our study is a first step in understanding how this methodology could be used. Certainly, as Alex points out, a 300 year averaging window is not necessarily useful in terms of available datasets. An important caveat that we will make much clearer in a revised paper is that the 300 year averaging window that we found was optimal does not necessarily represent a lower bound and this is something that certainly requires and warrants further investigation.*

*We have added an extensive new section in the paper that explain early warning theory, including a new figure. In addition, we have added much more discussion on the applications of this work in terms of ice sheet modelling and observations, as well as caveats to issues such as the window size that would be important to readers looking to apply this to their own work.*

2. You haven't necessarily explained why EWS show up in certain types of systems. To me, this is key to then explaining why you are calculating these things (ACF, variance, etc). In a canonical saddle-node bifurcation, we expect the stable eigenvalue of the linearized system state to smoothly decrease towards zero as you approach the bifurcation, which causes weaker damping of noisy forcing back towards the stable manifold. So, do EWS only occur where there is a saddle-node bifurcation? How do we know that MISI at PIG has such a smooth approach to the bifurcation? i.e. if the eigenvalue associated with the stable mode is controlled at first order by bed topography (which it is in the canonical formulation of MISI, see Schoof 2012 and others), then does the topography in the vicinity of the bifurcation need to vary smoothly towards a bedrock peak to produce EWS? Maybe these are questions for another study, but there needs to be some indication that you have grappled with the question of why you expect EWS to occur for PIG.

*Although EWS have largely been used to detect saddle node bifurcations these methods can be applied to other types of bifurcation (Scheffer, 2009), for example they have successfully been used in the context of Hopf bifurcations (Chisholm & Filotas, 2009). We will add this point to the revised paper. The second comment, regarding how bed roughness might affect the performance of EWS, is something we looked at during our modelling work but ended up not including in the paper submission. In fact, this was originally a major motivation of the PIG experiments because there was no guarantee that observing critical slowing in the idealised MISMIP experiment would work for a realistic topography. Before we did the PIG experiment we superimposed varying types and amounts of Perlin noise onto the smooth MISMIP bed, such that the overall retrograde bed slope became more obscured by smaller scale bed features. Repeating the same experiments on these new beds and extracting the relaxation time to predict the tipping point became successively less accurate as more noise was added. However even for a very bumpy bed there was still a trend towards longer relaxation times as the tipping point was approached and the*

*accuracy of the predicted tipping point was still good enough that we had confidence this approach would work for a real glacier. The resolution of both our PIG model and the bed topography are such that we resolve as much of this 'bumpiness' as possible in these simulations. Discussion around both of these points will be added into a revised version of the manuscript.*

*We have added further discussion in the manuscript around why early warning signals show up in certain systems and that this is not just true for saddle-node bifurcations. We have also added a section to the flowline experiment results describing the extended experiments we did whereby bed bumpiness was increased, and discussed the implications of this in the manuscript.*

3. Related to the issue above, more of the detail about how the EWS indicators are calculated would be helpful to bring into the main text, since this is a topic most TCD Interactive comment Printer-friendly version Discussion paper TC readers are not familiar with. How do you ensure statistical significance? Could we see EWS away from a bifurcation by chance? Why not?

*The issue of so-called 'false alarms', whereby an EWS is detected and incorrectly anticipates a bifurcation, is something that has been written about extensively in the literature and indeed it is entirely possible using this methodology. In our paper we do various statistics on the sensitivity and the significance of the indicators that we have calculated. These are currently in an appendix and would probably warrant being moved into the main text which we will do, along with reference to the possibility of false alarms.*

*The explanation of how early warning indicators are calculated has been expanded and is now consolidated into one part of the manuscript.*

4. You say that you force your model with variability from a "surrogate model" based on ocean variability in the Amundsen Sea region, but don't provide further details. First, more detail is needed on the surrogate model. Second, presumably this surrogate model produces ocean variability with significant power in the decadal range, as many studies (e.g. from Jenkins and others) have found that such variability is important in this region. However, the typical formulation of EWS (e.g. Lenton et al. 2008, and previous studies) assumes a martingale process for the noise forcing (i.e. white noise) which is not the case here. Can you explain why this doesn't affect your interpretations of EWS indicators? Third, it is unclear whether the steady-state and quasi-steady-state simulations used to make the bifurcation diagrams in Figure 3 include noise forcing. If not, then this is concerning, because it is well known that marine ice sheets have a different steady-state with and without noise in the forcing (e.g. Robel et al. 2018, Hoffman et al. 2019, Mikkelsen et al. 2018 (but for non-marine ice sheets)). This could be quite important in your simulations, since the location of the bifurcation is important to know for calculating EWS.

*The surrogate model was an AR based surrogate and does indeed contain a lot of decadal variability and this is important to capture as Alex and other authors have shown. We will provide more details on how this surrogate was made in a revised version of the paper. The second important benefit of adding this variability to our melt rate forcing is that, by perturbing the model, it helps to extract information on the system response time to identify critical slowing. It is true that the paper by Lenton 2008 shows results from earlier work by Held and Kleinen 2004 and in this case they used white noise with the same aim. The variability we add to our forcing is not white noise but there is no reason why this should affect our ability to extract information on the system response time. Indeed,*

*other authors have found critical slowing in a wide variety of paleo data (e.g. Dakos et al. 2008) and the natural variability that drove these systems is presumably also not white noise. The key to be able to detect EWS is to add some perturbation with a measurable impact on the system state and our surrogate time series succeeds in that with the added benefit of replicating natural variability as closely as possible. Indeed, using natural variability to test for EWS arguable has more relevance to this system than using white noise. With regards to our steady-state simulations (the squares and circles in figure 3b) no natural variability was added to these simulations. With the addition of strong variability in the forcing it is possible that some of these steady states might be nudged far enough that they end up in a different basin of attraction. However, we disagree with the reviewer that this is concerning. The location of the bifurcation for the purposes of searching for EWS was not defined based on these points but on the 'quasi steady state' points and so have no bearing on our results and merely serve to demonstrate that several hystereses exist. Having said that, it is more consistent to add the same variability in these runs. We have re-run these steady state simulations with the same natural variability that was added to the main PIG simulations and this has no significant impact on the location of these points but we will use these results, rather than the ones shown in this version of the paper, for a revised submission.*

*We have added a more detailed explanation of our surrogate time series and re-calculated all quasi-steady simulations to include the same variability as the transient simulations. This has not significantly affected any of the results and the figure now shows these steady state values instead of the previous ones that did not include variability.*

**Minor**:

Line 23: please define what you mean by "tipping element" for the uninitiated *This is now explained*

Line 29: grounding line flux *Done*

Line 47: the 2012 Schoof JFM paper makes more sense to cite here *Changed the citation*

Line 48: what do you mean by catastrophic?

*We use the word catastrophic in the sense that it is sometimes in literature on tipping points, that is a large change in the system state. We have made an effort to keep all terminology clear and consistent and since this is the only instance of that word in the paper we will rephrase it.*

*A wide variety of words are used in this context, another example is abrupt, but this implies a rapid response that is not necessarily true. We opted to call this a 'qualitative shift' to convey that the system is substantially different once it crosses a bifurcation point*

Line 89: So is accumulation held constant in time or does it have a seasonal cycle? Be more specific here, because its important to know if there is variability in more than just ocean melt.

*Accumulation is held constant in time, specifically to avoid adding variability in more than one forcing which could potentially muddy our results.*

*Clarified that accumulation is constant in time*

Line 119: Again, to be clear, it isn't necessarily the case that EWS exist for all tipping points (if we define any bifurcation as a tipping point).

*We addressed this comment above and will add a clarification on this point.*

*Added this clarification*

Line 126: it would help here to explain exactly how you force the simulations to produce the grey dashed and black lines in Figure 3 (also in caption). How fast is the forcing? How large are the step increments to determine the steady-states? How do you determine when there is a tipping point (just a large enough jump in the grounding line?)? What if you have a tipping point that causes the grounding line to retreat only a small amount?

*We will expand on this in a revised version of the manuscript.*

*The paragraph of line 126 is talking about the experiment in appendix A and not the one that generated the results in figure 3. We have added some of these details to the figure caption and the rest in the relevant section in the main body of text.*

Line 119-129: This whole paragraph is confusing

*Much of this paragraph has been rewritten*

Line 136: What do you mean by short tem "weather noise"? Isn't this the thing that is detected by EWS?

*This is a term that has been used in EWS literature to describe rapid but low amplitude fluctuations in the system state that do not represent quantities of interest*

*The wording has been changed to remove this term*

Line 184: What do you mean by "equivalent to a random walk"?

*A random walk describes an evolving system that consists of a succession of random steps but this is probably unnecessary jargon and will be removed.*

*This sentence is removed and instead we describe the DFA in more detail without this jargon in section 2*

Line 197-198: is this hysteresis related to the domain extent? If you aren't simulating flow from outside the domain, then when the whole domain collapses you won't be able to regrow the glacier for any parameter (because theres no catchment).

*The catchment is not completely gone and the grounding line remains within the basin but presumably you are correct that by extending the domain it might make it easier to regrow the glacier. That does not detract from our point in this case but we will include this as a possible explanation.*

*This point has been added*

Line 203-205: what about much shorter windows? (related to point 1 above related to observations). How short of a window would actually be calculable from observations (a decade? does this start to run into the AC time scale of forcing?)

*This is addressed above and will be discussed in more detail in the revised paper.*

*We have added substantial discussion on this point*

Line 214-215: this is a confusing sentence which leads me to think that you are saying the EWS are not actually "early". To what extent does this depend on the speed of the trend in forcing? Can you test it for different trend rates?

*We feel that the next sentence addresses this question to some extent but we can expand on that further. The EWS are identifying tipping in the transient simulation for which they are calculated, and this would be equivalent to the 'steady-state' simulations if the forcing was increased sufficiently slowly, but since the transient simulation evolves quicker than the glacier can adjust then these are not the same. To what extent the forcing rate affects various things is a potentially large and complex research question that we believe is beyond the scope of our study.*

*We have added discussion of this point in Sect. 5*

Figure 3a: If the black line doesn't fall on top of the gray dashed line, then the black line simulation isn't really quasi-steady. Why not call it something else? Also, it is unclear how the grey dashed line is determined, and which parts are stable and unstable?

*We will think of a word other than 'quasi steady' as this is possibly confusing. The dashed grey line is just a line through the individual model results represented by symbols, all symbols are steady states but it is possible that these are not true stable steady states.*

*We have clarified how the dashed grey line is determined in the figure caption. We discussed alternative names for this simulation but in the end we decided to continue with this name which is the same as has been used in similar studies on ice sheet hysteresis (e.g. the recently published paper by Garbe et al.) and instead we have added a better explanation defining what we mean by this.*

Figure 3b: please explain the different between "tipping point" and "instability onset". I can guess that the latter has to do with the region in which an unstable manifold exists (i.e. there is hysteresis), but I'm not sure readers will necessarily pick up on this without you explaining it explicitly.

*Instability onset is not the best choice of words here, and the legend and caption have been changed to EWI window to reflect that this is the portion of the time series that was analysed using early-warning indicators*

Figure 4: Looking at this (and Fig. B2), it seems clear to me that the length of time ahead of the tipping point needed to detect EWS is directly correlated to the speed of the forcing. Yet, that isn't really made clear here or in the text.

*The rate at which the forcing is changed is constant throughout the simulation (ignoring the added variability)*

*We have added discussion on the rate of forcing in Sect. 5*

Figure B2: can you extend this to shorter windows?

*Done*

Appendix A: Related to some of the issues I raised above, it may be valuable to bring the flowline simulations into the main text to demonstrate, in a very simple system where the exact location of the tipping points are known, how EWS work

*We have added extensive explanation of how early-warning signals work to address these concerns but have chosen to keep this section in the appendix to avoid the confusion of describing two very different model simulations with different goals in the same paper*

**Anonymous Referee #2**

In this manuscript the authors seek to detect the onset of marine ice sheet instability (MISI) in model simulations of Pine Island Glacier, using techniques that have previously been applied to other complex systems. The novelty of this study lies in the application of critical slowing indicators to confirm MISI events. It provides an interesting framework for evaluating vulnerabilities in ice sheets that will be of interest to the TC scientific community. However, particularly due to its novelty within glaciology, some aspects of the paper need improving to aid the clarity, and it would benefit from further exploration/discussion of the usefulness of the techniques beyond the modelling example provided here. I have outlined these below, followed by line-by-line comments.

*Many thanks for the positive evaluation of our study and the helpful comments. We replied to all comments and hope that our replies and changes made to the manuscript will address all issues raised. Our replies are in red and the changes to the manuscript summarised in blue.*

The paper includes a nice explanation and accompanying schematics of hysteresis; however, the critical slowing description and explanation of the indicators is less intuitive. This may partly be due to the structure – Appendix A offers a useful demonstration of how critical slowing manifests in a carefully controlled simple experiment. I think it is safe to assume most TC readers will not be familiar with these concepts, and therefore I do not think this example should necessarily be tucked away in an appendix. A diagram of critical slowing in a similar vein to Figure 1 would be helpful, or some additional annotation to Figure A1. There is a disconnect between the flowline example in Appendix A, and the methods used for determining the onset of a tipping point in the main set of experiments. Could you show how critical slowing in the flowline experiment can be demonstrated with the various indicators you use in your main experiments? This would help show how these indicators are related to the increased recovery time from a stepwise perturbation as the tipping point is approached.

The other reviewer also highlighted that various elements currently in the appendix could be moved into the main body of text and we will do so. We will also give a more detailed explanation of critical slowing since it is true that many TC readers will not be familiar with this concept and will produce a new figure in the introduction to hopefully aid this explanation. Our two sets of experiments, one using the MISMIP setup and the other using the PIG geometry, currently identify critical slowing in two different ways. This is intentional as these have different goals and we do not think it would be useful to directly compare them. The benefit of the MISMIP experiment is that it is simple and the way we extract the change in response time directly is very easy to understand. It is also an experiment that many readers of TC and the wider glaciological community are very familiar with. However, since the bed is smooth and the perturbations to the model are stepwise, we can not detect EWS in the flowline setup using the same approach as was used for the PIG experiment.

*We have added a new section to the paper that gives a more detailed explanation of critical slowing and early warning indicators, including a new figure that explains these concepts diagrammatically. We have added further explanation to the differences between the flowline results and main results to clarify why the indicators could not be used in the flowline case.*

My other main comment is about the usefulness of this method in detecting EWS in reality. Would the 300-year optimal window size apply to other catchments? What kind of observational datasets are required to implement this analysis, in a way that would act as a useful EWS for MISI? The measurement used in this study (grounding line flux) does not exist (at least not at the quality/resolution required here) prior to the satellite era, so what is the alternative, given the 300-

year window size? In your model simulations the forcing is applied gradually in order to avoid "one tipping point cascading into the next and result in three individual tipping points being misinterpreted as only one event" (L168). What are the implications of this for detecting tipping points in observations, where the system is not necessarily able to return to a quasi-steady-state with changing forcings? Do you have a sense of whether the indicators would hold up if the forcing is more rapid? Further discussion of these issues would strengthen the paper.

*Many of these points are raised by reviewer 1 and we have addressed them at length in that reply. To summarise our response: we will add more discussion around the implications of our results for finding EWS in observational records. We do not believe the 300-year window size to be a lower bound and did not try to reduce this systematically, but this will also be discussed in a revised version of the paper. Since the length of record necessary for EWS as well as the variable used to detect EWS are critical parts of what observations can or cannot be used, this is a very important research question but one that requires considerable work to address and is beyond the scope of this initial study. Regarding the last point about how cascading tipping points and a rapid forcing might affect our ability to detect critical slowing, this is known to influence the ability to detect EWS (Dakos et al., 2015). For example in ecosystems, interacting regime shifts can muffle or magnify variance near critical transitions (Brook and Carpenter, 2010). In this study, as explained in detail in the response to reviewer 1, we first of all want to make the nontrivial case, that ESW – that exist theoretically for MISI – can actually be detected in a realistic geometry that is of great interest with respect to MISI at the moment. We would argue that this issue raised by the reviewer is very important and an interesting research questions but beyond the scope of this study.*

*We have added extensively to the manuscript to discuss all the points raised here by the reviewer: the usefulness of the method, the window size, observational datasets required and cascading tipping points.*

Other comments:

L13: "Self-amplifying retreat" this could be considered an overstatement. Selfsustaining retreat would be more accurate (and more in keeping with language further on in the manuscript).

*Changed as suggested*

L18: "early warning indicators robustly detect critical slowing for the marine ice sheet instability". It might be worth removing the term "critical slowing" from the abstract, and instead using a less jargon-y alternative, e.g. "robustly detect the onset of MISI"?

*Changed as suggested*

L31-32: "a complex range of factors can either cause or suppress the MISI" – such as? The two papers cited refer to buttressing, what about local sea level, GIA etc? Haseloff, 2018, should be Haseloff and Sergienko, 2018.

*Further citations added for other factors*

L68-70: "Our results reveal the existence of multiple smaller tipping points that when crossed could easily be mis-identified as simply periods of rapid retreat, with the irreversible and the self-sustained aspect of the retreat being missed": this seems to contradict your results and conclusions. The two smaller tipping points are not irreversible, as the system can return to previous state through stronger perturbations in the opposite direction – shown by the hysteresis loops in Figure 3.

*Unhelpfully, the word irreversible is used by various authors to mean different things in this context and we have tried to be consistent throughout the paper and we clarify our meaning in lines 53-54. We are not aware of a word that uniquely describes one or other type of irreversibility but this is an important point to clarify.*

*We now differentiate between the two types of behaviour here and throughout the manuscript*

L106: Basal melt rates: it is not clear from this paragraph whether basal melt occurs under grounded ice. How do you treat partially grounded elements? This has been shown to be important in modelling grounding line retreat (e.g. Seroussi and Morlighem, 2018, doi: 10.5194/tc-12-3085-2018).

*Melting is only ever applied to fully floating elements.*

*This point is added to the revised manuscript*

L153-159: Unlike the other paragraphs in this section, this paragraph does not contain an outcome of your decision-making process – which of the criteria will you use?

*We find that indicators do seem to reach critical values ahead of a tipping point and but this is a result of our study rather than a decision made during the experiment design.*

*This paragraph has been rewritten to clarify how we present our results and why*

L180-181: "Furthermore, the indicator reaches a critical value relatively close in time to when the MISI event gets underway". Clarify that the critical value is 1.

*Done*

L183-185: "For this early warning indicator. . .". I don't understand this sentence and it seems like it would be better suited (with added detail) to the methods. I thought both indicators have a critical value of 1 (section 2.2), so why does scaling the DFA help with comparison to ACF?

*The DFA indicator does not have a critical value of 1, hence why scaling it makes sense.*

*This is hopefully now much clearer as a result of the more detailed explanation of the DFA indicator*

L188-189: "although variance cannot be used directly to predict when that threshold will be crossed" – perhaps this is obvious, but why not? Because there is no critical value? But crossing the critical value doesn't seem like a robust way of detecting the exact onset of MISI either, considering some of the trend lines in Fig. 4 cross x=0 before they reach the critical value?

*Yes, because variance has no critical value. It is true that our trend lines do not cross critical values exactly at the tipping point, that would be remarkable given the complexity of the model, however we do find our indicators reach critical values very close to a tipping point, making this a useful EWS.*

*We have clarified why variance cannot be used and also why it is useful later in the discussion*

L275: What do you mean by "i.e. a record length of 600 years" – how does that relate to the window size? Is that the minimum record length required?

*This point is addressed at length elsewhere in our response.*

L258: Basin of attraction has not been defined/explained.

*Reworded this sentence*

Figure A1: Panel A, grounding line position in km: clarify the direction of retreat.

*This is now annotated on the figure*

**The tipping points and early-warning indicators for Pine Island Glacier, West Antarctica**

S.Sebastian H. R. Rosier[1*], R.Ronja Reese[2], J.Jonathan F. Donges[2,3], J.Jan De Rydt[1], G. H.Hilmar Gudmundsson[1], R.
[revised manuscript text omitted]

---

## Editor Decision (ED1)

Dear authors,

Thank you for addressing the points raised by Reviewer 1 in their second review of your article on "The tipping points and early-warning indicators for Pine Island Glacier, West Antarctica". This is a very clear article on an important emerging area of research. I list below a number of small technical corrections that should be addressed prior to publication but otherwise this article can now be published in The Cryosphere.

Kind regards,

Pippa Whitehouse

To be corrected (all line numbers refer to the non-track change version of the article):

Line 17: 'WAIS' – acronym not defined

Line 25: 'sea-level rise' should always be hyphenated. Please also check that use of a hyphen in the phrase 'early warning' is consistent throughout the text

Line 34: update the Oppenheimer et al. reference

Line 70: PIG is already defined on line 37

Line 194: pacific -> Pacific

Line 422: the relaxation time *increases* as you approach the tipping point ($T_R^{-2}$ decreases)

Line 731: please use consistent units for basal melt rate

Suggested edits:

Line 14: 'committing a glacier to…' – MISI theory was developed to describe ice sheet behaviour but can be applied to glaciers under certain conditions. Suggesting revising the text to reflect the original purpose of the theory

Lines 23-24: sentence revised in response to reviewer comments, but grammar is now awkward

Line 30: do you mean 'increase in accumulation'?

Line 76: 'multiple smaller tipping points' – smaller than what? Are these in addition to the main three tipping points identified by your analysis?

Lines 80-81: in your response to reviewer 1 you explain the difference between early warning signals and early warning indicators; it would be useful to also summarise the difference for the reader

Line 117: 'to be more similar' -> 'to become increasingly similar'

Line 129: details of the rescaling are unclear; you do not define what you mean by 'a critical value' and it is not clear what value 0.5 (white noise) is mapped to

Line 274: 'a range of melt rates between these two states' – it is implied, but not explicitly stated, that each steady state can be related to a specific melt rate, please clarify

Line 295: 'We show results' – make it clear that this phrase relates to results presented above

Line 340: 'may have failed' -> 'may fail'

Line 361: 'this methodology' -> 'our methodology'

Line 368: delete 'that we did not identify'

Line 402: 'a +1.2˚C change in ocean temperatures' – relative to what?

Figure 4 caption: 'The steady states… are plotted as dashed grey lines' – rephrase to say that the steady states plot along the grey dashed lines, and the details are shown in Fig. 4b

---

## Author Response (AR2)

We would like to thank Alex Robel for his thoughtful and helpful comments on our manuscript which have led to considerable improvements from our original submission. Alex's latest comments are given below in bold, with our response to the comments together with the changes made to the manuscript are given in blue.

**Overall, this revised manuscript provides an excellent proof of concept for using Early Warning Indicators (EWI) as a tool for diagnosing the Marine Ice Sheet Instability in ice sheet model simulations. The revision now also serves as an excellent primer on the theory and practice of calculating EWIs that will be useful to those in the glaciology community who are new to this topic. My overall recommendation is that this paper is now close to being ready for publication, with a few more suggestions, which could be classified as "minor".**

**My more substantive suggestions:**
**1. In my original review I was fairly convinced that this study primary utility was as a proof of concept for showing how EWI could be identified observationally for MISI. However, in this version, it isn't until lines 352-354 that it became clear to me the exact utility of using EWI-type analysis on simulation output, beyond as a proof of concept. Here the authors state that this approach could be useful to lower the computational expense needed to verify the presence of bifurcations in a model simulations (i.e. without very long simulations). As a modeler, I find this to be a compelling argument, and should be front and center in the paper, particularly in the first section where you are trying to entice readers to continue learning more about EWIs.**

We agree that this is an important potential application of the EWI and Alex makes an excellent point that this is not mentioned until very late in the manuscript. We have added several sentences covering this point in the second paragraph of the introduction.

**2. Not to hammer too hard on the window length issue, but I think there needs to be a bit more discussion of two points.**

**(a) First off, on lines 280-282, you state 300 years is "the shortest window size for which the DFA indicator provides an accurate prediction for all tipping event". What does "accurate" mean in this context? From Figure 7, I would guess you mean where Kendall's Tau has a maximum, but why does that make this window the most accurate way to determine whether a tipping point will occur? In principal, wherever $0<\tau<1$ is indicative of increasing indicator, so one could argue that window lengths anywhere from 200 to 400/500 years seem to work in this regard.**

This statement is admittedly rather strongly worded since it is true that a positive tau coefficient was obtained for smaller window sizes. Our intended meaning was simply that below ~300 years for all indicators and tipping point events (but closer to ~200 years) for some, the tau coefficient becomes less strongly positive, indicating a less clear increase in each indicator before the tipping point. This in practice would represent less certainty in early warning detection. We have reworded this and also the section that introduces the Kendall's tau coefficient so together these changes hopefully make our intended meaning clearer.

**(b) There is also still not much investigation of window lengths relevant to observational time scales (i.e. decades). I wonder if increasing the forcing rate to levels that might be relevant to modern climate change (i.e. degrees/century) would make short window lengths more useful? This is perhaps beyond the scope of the paper, but if its computationally feasible to run a few simulations with higher forcing rate, and re-calculate figure 7 for those forcing rates, you could**

**begin to answer the questions of whether it would be possible to use modern observations to calculate EWIs.**

We agree with Alex that this is something very interesting to look at in the future but think that it is beyond the scope of the current paper. This question of what determines a 'minimum' practical record length to apply EWIs to observations is a study in and of itself and would require a large number of computationally expensive simulations to address properly – not just by exploring how it is affected by window size but no doubt other factors such as the frequency of variability in the forcing are highly relevant. The choices that we have made in this paper represent carefully selected values and given the nature of this paper as a first step in applying EWIs to ice sheet modelling, and the significant delays to publication that these lengthy simulations would involve, we prefer to reserve this suggestion for future studies.

**3. In my opinion, the point that is made throughout about separating tipping points by using slow forcing is a bit off the mark. Because you are not doing fully steady-state simulations, it is not clear that you have shown that these are actually three discrete tipping points (in the traditional sense of a bifurcation diagram which traces the stable manifold of the system). In the real system, these three tipping points likely involve a grounding line retreating over a region of reverse sloping bed with some prograde bumps. Perhaps if the forcing were even slower, there would be places where the grounding line stabilizes on these bumps, subdividing these tipping points even further. On the other hand, for realistically fast forcing rates, you would have a "tipping point cascade" that might look like one single rapid retreat. This would still be of interest, and could be valuable to identify using EWI since it is closer to what we are likely to see in reality. This is all to say that without a completely steady-state analysis it seems a bit premature to argue that you have found the three actual tipping points in the system, when there may be more in a mathematical sense, or when they might combine into one tipping event under real forcing.**

We agree with many of these points and they are largely covered in the manuscript already. We state at the start of our results that 'We focus our results on these three major changes in the glacier configuration and ignore any possible smaller tipping points that do not result in significant grounding line retreat or changes in ice volume'. We do not claim anywhere that there are only three tipping points but state that three distinct tipping points can be identified from our model runs. While it is true that we first identify 'potential' tipping points using transient calculations, but we then go on to do fully steady-state simulations to verify that these are tipping points in Section 4.2. It is also true that a faster forcing might cause all three to be crossed at once and we cover this point in the manuscript. Finally, we agree that it is possible that a slower forcing, or alternatively using a smaller interval in the control parameter for our steady-state simulations, might identify additional potential tipping points. We have altered our wording to better reflect that we have not necessarily identified every single tipping point in a mathematical sense and added a sentence to the discussion on this point. In general, however, we chose to continue to refer to three tipping points because we are very confident that our methodology detected the three largest and most societally relevant tipping points in our model simulations.

**Minor issues:**
**Line 12: what does "this" refer to in this sentence?** Replaced 'this' with 'ongoing and future changes'

**Line 18: indicators in model simulations robustly detect** Done

**Line 23: delete "a major component of the earth system" to make sentence clearer** This part of the sentence was originally added in response to a previous request to clarify what is meant by a 'tipping element' but we have changed the sentence to hopefully make it read better.

**Line 27: If grounding line retreat causes grounding line flux to increase** Done

**Line 31: a small perturbation results in the system** Done

**Line 43: delete "externally forced" since there needs to be an external forcing trigger for MISI to occur in the first place** Done

**Line 47: I'm not sure you need to mention the lower stable branch since technically it does not "participate" in the bifurcation (and there doesn't need to be a lower stable branch at all to have a saddle node bifurcation)** The lower stable branch is shown in figure 1 and as is commonly done with diagrams of this type and so we prefer to keep this description.

**Line 46-55: It would be useful to indicate what are the assumptions under which it is the case that MISI is a saddle-node bifurcation? i.e. that bed slope is negligible and changes very slowly in space (i.e. Schoof 2007/2012)** These are the assumptions used by Schoof but do not necessarily represent necessary assumptions in general for the MISI to be a saddle-note bifurcation. We show that the MISI tipping point behaves as a saddle-node bifurcation but these assumptions are not made in any of our simulations that use a realistic geometry.

**Line 51: parameter range** Done

**Line 58 and elsewhere: I always thought this was called "critical slowing down". A cursory search in the literature indicates that this is the most common usage and should perhaps be used that way here if you want readers to relate this to the broader EWS literature.** Replaced all instances with critical slowing down

**Line 76: are you calling these EWI or EWS? You use both in the same sentence** We believe this usage is correct: the early warning signals manifest in the data or model output e.g. critical slowing down but this are analysed with the use of early warning indicators e.g. lag-1 autocorrelation.

**Section 2.1: So, I am typically loathe to reference my own papers, but I think it bears noting that Robel et al. 2018 shows analytically that the response time (calculated directly from the eigenvalues of the system) increases towards the MISI bifurcation (see Fig. 3 in that paper).** This paper is certainly directly relevant and was missed in a literature search so we are grateful to Alex for pointing it out and a sentence has been added to section 2.1 on this point with the reference added.

**Line 123-124: This sentence could be written a bit more clearly since its unclear what you are saying about variance here** This explanation has been expanded

**Line 154: typo at exponent** Done

**Line 191: Also, by using realistic noise you can assess EWI detectability that would be expected in observations** Yes indeed, added this point

**Line 199: How do you know what is not related to system recovery time? Sentence is a bit vague** Reworded this to make it clearer what we mean

**Line 220-224: can you be clearer about the different interpretations of tau=1 and 0<tau<1?** We have rewritten this section slightly to hopefully make this clearer

**Line 247: MISI event begins** Done

**Line 314: of ice flow are** Done

**Line 337: and also decreasing window length?** True but this is mentioned elsewhere, the point here is specifically that increasing distance from the tipping point eventually reduces the predictive power of EWIs

**Line 346: related to issue #3 above, but it isn't clear why this cascade is a problem from the POV of EWI detectability** We have considered this further and realise some confusion may arise from our using of the term 'tipping cascade' which has been used with various meanings in the past. Perhaps the stricter meaning is one tipping point causing a second tipping point to be crossed and in that sense this is not precisely what we are referring to here. A tipping point can be crossed either through changes in the system state or the control parameter. However, these processes are not instantaneous and if the control parameter is changing sufficiently quickly then it could trigger a second tipping point to happen soon after the first in a way that might not have happened if it were held constant or changed very slowly after the first tipping point was crossed. We called this a tipping cascade in the sense that one tipping point 'cascades' into the next but it is a different mechanism than the definition above. That being said, cascading tipping points are also an issue since they would complicate any interpretation further and the methodology we use whereby a control parameter changes continuously with time could not distinguish between one tipping point or two cascading tipping points. We have changed the wording here to make our meaning clearer.

**Line 349: tipping points is one that** Done

**Figure 4: in panel (b) you have open markers where you have done steady-state calculations, but not in panel (a), so it is unclear where you have actually done simulations to determine steady-**

**states.** This is done just to improve the readability of panel a and avoid clutter and the last two sentences of the figure caption explains where steady-state calculations were done but we have expanded on this explanation to make it clearer.

---

## Author Response (AR3)

Thank you to the editor for going through the paper once more in detail and spotting a number of mistakes. Your comments are in italics and our response to each comment in bold.

*Line 17: 'WAIS' – acronym not defined* **Done**

*Line 25: 'sea-level rise' should always be hyphenated. Please also check that use of a hyphen in the phrase 'early warning' is consistent throughout the text* **All instances of hyphenation in early warning and sea-level rise fixed**

*Line 34: update the Oppenheimer et al. reference* **Done**

*Line 70: PIG is already defined on line 37* **Done**

*Line 194: pacific -> Pacific* **Done**

*Line 422: the relaxation time increases as you approach the tipping point (TR -2 decreases)* **Done**

*Line 731: please use consistent units for basal melt rate* **Done**

Suggested edits:

*Line 14: 'committing a glacier to…' – MISI theory was developed to describe ice sheet behaviour but can be applied to glaciers under certain conditions. Suggesting revising the text to reflect the original purpose of the theory* **The difference is largely semantic and although frequently associated with an idealised ice sheet configuration in reality the theory is most relevant to individual glaciers since these are the dynamic portions of the ice sheet where grounding lines will be advancing/retreating. It is true that Weertman originally referred to ice sheets in his 1974 paper, but that is not a limit on the theory and what is important are the mathematical assumptions on which the theory is based, most notably of course that it is two dimensional. Since we do give an overview of MISI theory in the first paragraph and this association with glacier retreat is very common we do not feel that it would lead to confusion regarding the important aspects of MISI theory and referring to glaciers in our case makes sense since this is what we are modelling.**

*Lines 23-24: sentence revised in response to reviewer comments, but grammar is now awkward* **Done**

*Line 30: do you mean 'increase in accumulation'?* **Yes indeed, changed as suggested**

*Line 76: 'multiple smaller tipping points' – smaller than what? Are these in addition to the main three tipping points identified by your analysis?* **Here we refer to the three tipping points identified in the paper and we said smaller to reflect that several tipping events led to a collapse of the glacier, rather than one single event. Wording altered to make this point clearer.**

*Lines 80-81: in your response to reviewer 1 you explain the difference between early warning signals and early warning indicators; it would be useful to also summarise the difference for the reader* **Done as suggested**

*Line 117: 'to be more similar' -> 'to become increasingly similar'* **Done**

*Line 129: details of the rescaling are unclear; you do not define what you mean by 'a critical value' and it is not clear what value 0.5 (white noise) is mapped to* **We added a sentence defining what is meant by a critical value. We now explicitly refer the reader to the details of the rescaling in Livina and Lenton (2007)**

*Line 274: 'a range of melt rates between these two states' – it is implied, but not explicitly stated, that each steady state can be related to a specific melt rate, please clarify* **Reworded this sentence**

Line 295: 'We show results' – make it clear that this phrase relates to results presented above **Done**

Line 340: 'may have failed' -> 'may fail' **Done**

Line 361: 'this methodology' -> 'our methodology' **Done**

Line 368: delete 'that we did not identify' **Done**

Line 402: 'a +1.2°C change in ocean temperatures' – relative to what? **Specified this is relative to initial model conditions**

Figure 4 caption: 'The steady states… are plotted as dashed grey lines' – rephrase to say that the steady states plot along the grey dashed lines, and the details are shown in Fig. 4b **Done**